# Graph-Theoretic Intrinsic Reward: Guiding RL with Effective Resistance

**Jatin Chauhan**[†]**, Shivam Bhardwaj**[†]**, Aditya Saibewar, Aditya Ramesh,**
**Sadbhavana Babar** & **Manohar Kaul**
Fujitsu Research *

## Abstract

Exploration of dynamic environments with sparse rewards is a significant challenge in Reinforcement Learning, often leading to inefficient exploration and brittle policies. To address this, we introduce a novel graph-based intrinsic reward using Effective Resistance, a metric from spectral graph theory. This reward formulation guides the agent to seek configurations that are directly correlated to successful goal reaching states. We provide theoretical guarantees, proving that our method not only learns a robust policy but also achieves faster convergence by serving as a variance reduction baseline to the standard discounted reward formulation. We perform extensive empirical analysis across several challenging environments to demonstrate that our approach significantly outperforms state-of-the-art baselines, demonstrating improvements of up to 59% in success rate, 56% in timesteps taken to reach the goal, and 4 times more accumulated reward. We augment all of the supporting lemmas and theoretically motivated hyperparameter choices with corresponding experiments.

## 1 Introduction

Training policies in sparse-reward settings has been a longstanding challenge, which has now become mainstream with increasing deployments of autonomous agents in real world. Initial series of works focused on leveraging meticulously engineered dense reward functions to guide the learning process (Schulman et al., 2017; Barto, 2021), which achieved success in tasks ranging from simple pathfinding to complex motor control (Schulman et al., 2017; Barto, 2021). However, the reliance on such dense rewards inherently limits the scalability (Cao et al., 2024; Antonyshyn & Givigi, 2024). On the contrary, in the sparse reward setting the agent receives a positive signal only upon reaching a distant goal, with no intermediate feedback to guide its exploration. This credit assignment problem often renders simple exploration strategies, such as $\epsilon$-greedy action selection or unstructured noise, rather ineffective as the agent is unlikely to encounter the reward achieving state through random behavior (Pitis et al., 2020). To overcome this, intrinsic motivation based methods have garnered much attention and success recently (Achiam & Sastry, 2017; Xudong et al., 2024), which typically generate a dense reward signal to encourage meaningful exploration. These signals are often based on information-theoretic concepts like novelty, curiosity, or surprise (Achiam & Sastry, 2017; Pathak et al., 2017; Burda et al., 2018). These methods however have caveats. Most of these methods provide empirical validation, however fail to justify the algorithms with theoretical guarantees. Furthermore, methods like (Xudong et al., 2024), albeit better than their predecessors like Hindsight Experience Replay (HER) (Andrychowicz et al., 2018) and MEGA (Pitis et al., 2020), further require policies pretrained via Behavioral Cloning to operate well, which can be expensive in many situations. Other methods such as the Surprise Based notion of (Achiam & Sastry, 2017) focus on model-based RL where the transition probabilities are modeled explicitly, which is highly non-trivial (Viano et al., 2021).

Accounting for all these factors, we introduce a novel principled approach to intrinsic motivation for goal-conditioned RL with roots in spectral graph theory. We model the agent's perception of its dynamic environment, where all entities move arbitrarily, as a graph that evolves over time.

---

*† denotes Equal Contribution. Email Correspondence: {*chauhan.jatin, shivam.bhardwaj, saibewar.aditya, aditya.ramesh, sadbhavana.babar, kaul.manohar*}@fujitsu.com*

The nodes of this graph characterize all the objects (including the agent and the goal) whereas the weighted edges describe the vicinity relative to the agent's sensors. As the primary contribution of this work, we propose to use the `Effective Resistance` between the agent and goal nodes as intrinsic reward. Intuitively, effective resistance quantifies the information flow between two nodes in a network, considering all possible paths between them (Evans & Francis, 2021). These possible paths naturally account for all the objects present within the agent's sensing radii. A decrease in this resistance signifies that the goal has become more structurally accessible within the agent's perceived map that may not be quantified by using simpler metrics such as Euclidean distance.

We emphasize that our proposed formulation is broadly applicable to goal-conditioned RL, where the observable (fully or *partially*) state space can be decomposed to extract the entities of interest in order to construct any graph. We provide detailed discussion of such settings in section A.14. Furthermore, we focus on sparse and constrained environments since such settings serve as more interesting and challenging test cases in real world. We demonstrate this via an illustration in figure 1 where the agent to goal relative distance remains same but the second configurations makes the task success much more feasible by providing better pathways to the agent. In both cases, evolution of Euclidean distance cannot provide any meaningful signal whereas the change in Effective Resistance provides a dense reward signal to update the policy. Furthermore, contrary to the inefficiencies of the aforementioned methods in the literature, our method does not require any pre-training and the whole process is on-policy. We also do not restrict ourselves to explicit model-based formulation either, thus generalising our contribution.

We provide various theoretical results for the time evolution of the effective resistance ($\mathcal{R}_{\text{eff}}$) in correspondence to the state configuration *relative* to the agent. This translates to the evolution of graph connectivity (generated over the state) showing that as the $\mathcal{R}_{\text{eff}}$ reduces, the connectivity improves. We also characterize various lemmas in section 4 on its effect on the gradients of the policy and improved goal visibility. Additionally, we show that our formulation serves as a variance reduction baseline leading to improved sample complexity and faster convergence.

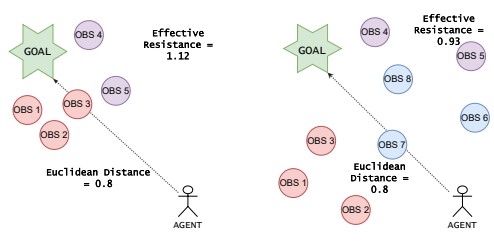

Figure 1: An illustrative example showing two configurations - (a) first where the agent's path is bottlenecked resulting in a denser local graph (section 3.2) and a higher effective resistance value identifying it as a less sought after configuration, (b) second with lower Effective Resistance where the agent can navigate through various pathways. In both cases, the Euclidean distance remains same and the direct path to the goal is obstructed, thus any such metric does not provide meaningful signal.

Our contributions in this work are thus fourfold. First, we formalize the use of effective resistance as a surprise-based intrinsic reward for goal-conditioned reinforcement learning. Second, we provide theoretical guarantees for our method, proving that it leads to robust navigation policies and faster convergence. Third, we conduct extensive evaluations on a suite of challenging navigation tasks with sparse rewards from the Safety Gym library by (Ji et al., 2024), outperforming state-of-the-art baselines by upto $59\%$ in success rate in achieving the goal, $56\%$ in timesteps taken to achieve the goal as the trajectories unfold, and $4\times$ better reward. Fourth, we provide a detailed and rigorous empirical validation of all theoretical claims, especially including the principled selection of key hyperparameters.

## 2   RELATED WORK

Our work builds upon two primary pillars of reinforcement learning research: goal-conditioned reinforcement learning (GCRL) or goal-conditioned policy optimization (GCPO) and intrinsic motivation for exploration. GCRL extends the standard RL framework by training policies that can generalize across a wide range of goals (Liu et al., 2022; Nasiriany et al., 2019), making it a core component of hierarchical agents and unsupervised skill discovery (Pitis et al., 2020). A key enabler for GCRL in sparse-reward settings has been Hindsight Experience Replay (HER) (Andrychowicz

et al., 2017), which relabels failed trajectories to create dense learning signals. While effective, HER-based methods are predominantly off-policy and struggle with certain task structures, such as those with non-Markovian rewards where a single state cannot be treated as a hindsight goal (Xu et al., 2025). (Xudong et al., 2024) recently proposed a formulation for trajectory selection that improves over previous works in this setting. They also provide a detailed summary of this line of work and as a recent SOTA method, we have used it as a baseline in our work. Our on-policy approach circumvents this limitation while still addressing the core challenge of exploration in multi-goal settings.

To tackle the exploration problem in sparse-reward tasks, researchers have proposed various forms of intrinsic motivation to generate dense rewards. These methods encourage an agent to explore based on information-theoretic concepts like novelty, learning progress, or surprise (Achiam & Sastry, 2017; Pathak et al., 2017). Surprise, in particular, has been formulated as the divergence between an agent's learned world model and the true environment dynamics, incentivizing the agent to visit unfamiliar states (Burda et al., 2018). Another line of work focuses on maximizing the entropy of the achieved state or goal distribution, encouraging broad exploration of the agent's capabilities (Pitis et al., 2020). Unlike methods that model transition probabilities directly (Achiam & Sastry, 2017; Liu et al., 2022) or sample goals from a learned density such as (Pitis et al., 2020), our approach uses the structural properties of a dynamically generated graph from the state configuration relative to the agent in order to create an intrinsic reward that naturally for all possible pathways and all instances in the environment.

## 3 METHODOLOGY

### 3.1 PRELIMINARIES

**MDP formulation:** We model the task as MDP defined by the tuple $\mathcal{M} = (\mathbf{S}, \mathbf{A}, \mathbb{P}, R, \gamma)$. The state space $\mathbf{S}$ consists of observations $s_t \in \mathbb{R}^N$, with the actions $a_t \in \mathbb{R}^{\hat{N}}$ (see section A.2 for details). The transition probabilities, $\mathbb{P}(s_{t+1}|s_t, a_t)$, are unknown, which is the case for majority of practical environments. The reward function $R(s_t, a_t)$ is a composition of an extrinsic and an intrinsic signal with $\gamma$ being the discount factor. The extrinsic reward, $r_{\text{ext}}(t)$, is *sparse*, providing a positive reward *only for reaching the goal* (which may never happen (Vasan et al., 2024; Hare, 2019)) along with a negative cost for situations such as collision with other dynamic objects in the environment. The intrinsic reward, $r_{\text{int}}(t)$, is a dense signal we *design* to guide the agent. The total reward at time $t$ is a combination of both with a weighting hyperparameter $\alpha$

$$r_{\text{total}}(t) = r_{\text{ext}}(t) + \alpha \cdot r_{\text{int}}(t) \tag{1}$$

**Problem Statement:** We use the standard objective in RL, where the aim is to find the optimal parameters $\theta$ for the policy $\pi_\theta(a_t|s_t)$ that maximize the expected discounted reward $\theta^* = \arg\max_\theta \mathbb{E}_{\zeta \sim \pi_\theta} \left[ \sum_{t=0}^T \gamma^t r_{\text{total}}(t) \right]$ where the trajectory $\zeta = (s_0, a_0, s_1, a_1, \dots)$ is generated by executing the policy $\pi_\theta$. The core challenge lies in the sparsity of $r_{\text{ext}}(t)$, which makes exploration difficult and renders standard algorithms ineffective as shown in section 5.

### 3.2 INTRINSIC REWARD VIA EFFECTIVE RESISTANCE

Intrinsic rewards have been shown to encourage learning, proving especially beneficial for environments with sparse rewards (Devidze et al., 2022). Furthermore, the combination of intrinsic and (sparse) extrinsic rewards improves overall learning by balancing such exploration behavior and task-specific goal achievement during the policy update (Zheng et al., 2024).

We ground our intrinsic reward in *Effective Resistance* (Chung, 1997). Unlike trivial metrics such as Euclidean distance which doesn't provide an effective signal as discussed earlier, the brittle Shortest Path Distance (Goldberg & Harrelson, 2005) which is susceptible to single-point failures or coarse global measures like Algebraic Connectivity (Fiedler, 1973), effective resistance provides a goal-oriented, pairwise metric that holistically considers the all available paths. This property yields a dense signal that promotes reliable navigation as we have proved in Theorem 1 and validated empirically in section 5. We now discuss the graph construction from the state vector $s_t$ needed to compute this metric.

**Graph Construction:** A key component of our methodology is the transformation of the high-dimensional state vector $s_t$ into a weighted, undirected time evolving graph $\mathcal{G}_t = (V_t, E_t, W_t)$. The state vector $s_t$ characterizes the environment's topology in a localized neighborhood w.r.t to the agent's (such as LiDAR), providing a very broadly applicable egocentric framework (Liu et al., 2023b). Additionally, other variants of graph formulation from the state vector $s_t$ have also been shown to work remarkably well for some RL settings such as DQN (Waradpande et al., 2021).

In $\mathcal{G}_t$, the vertex set $V_t$ includes nodes representing the agent, the goal, and distinct objects or object clusters. The edge set $E_t$ and their corresponding weights $W_t$ represent the connectivity and proximity between these entities. Our graph construction algorithm consists of the following steps - (i) construction of nodes of similar object categories (clusters) with a separate single node for the agent (egocentric formulation) in line 6, (ii) connecting the agent node to the remaining nodes in line 11, (iii) introducing intra and selective inter-cluster connectivity (in line 24). The complete details are provided in Algorithm 1. The specific design choices for this graph construction (Algorithm 1) are not arbitrary and we provide a detailed rationale in Appendix A.5.1, an analytical derivation for the clustering threshold $\tau$ in Appendix A.5.3 and a full empirical sensitivity analysis in Appendix A.9.

**Effective Resistance:** For two nodes $u, v \in V_t$, effective resistance measures the potential difference between them when a unit of current is injected at $u$ and removed at $v$. It is computed via the Moore-Penrose pseudoinverse, $L_t^+$, of the graph's Laplacian matrix $L_t$:

$$\mathcal{R}_{\text{eff}}(u, v; \mathcal{G}_t) = (e_u - e_v)^T L_t^+ (e_u - e_v) \tag{2}$$

where $e_u$ is the standard basis vector for node $u$. We refer to $\mathcal{R}_{\text{eff}}(u, v; \mathcal{G}_t)$ as $\mathcal{R}_{\text{eff}}(t)$ hereon. We define the agent's intrinsic reward based on the change in effective resistance between the agent node $\mathcal{A}$ and the goal node $g$. Let $\mathbf{1}_{\text{goal}}(t)$ be the indicator function for goal being present in agent's sensing radii and thereby its node in $V_t$. The intrinsic reward $r_{\text{int}}(t)$ is defined as:

$$r_{\text{int}}(t) = \begin{cases} -(\mathcal{R}_{\text{eff}}(\mathcal{A}, g; \mathcal{G}_{t+1}) - \mathcal{R}_{\text{eff}}(\mathcal{A}, g; \mathcal{G}_t)) & \text{if } \mathbf{1}_{\text{goal}}(t+1) = \mathbf{1}_{\text{goal}}(t) = 1 \\ -\beta & \text{if } \mathbf{1}_{\text{goal}}(t) = 1, \mathbf{1}_{\text{goal}}(t+1) = 0 \\ +\beta & \text{if } \mathbf{1}_{\text{goal}}(t) = 0, \mathbf{1}_{\text{goal}}(t+1) = 1 \\ 0 & \text{otherwise} \end{cases}$$

where $\beta \gg 0$ is a hyperparameter penalizing goal loss and rewarding goal recovery. Since our $r_{int}$ formulation is generic, it can be augmented with any standard RL algorithm (we use PPO in our case, see section A.5.2 for all the experimental details). We highlight that the graph construction in our method is focused on guiding the reward formulation, whereas the actions are predicted using $s_t$ as input to $\pi_\theta(a_t | s_t)$.

We distinguish our analytic formulation from recent approaches in Quasimetric Learning (Wang et al., 2023; Liu et al., 2024), which learn distance metrics from interaction data. While such methods offer flexibility, we emphasize that our effective resistance-based approach leverages the strong inductive bias of spectral graph theory. This grants our method immediate structural interpretability and theoretical guarantees without the high sample complexity required to learn a metric from scratch. We provide a detailed comparison of these paradigms in Appendix A.17.

## 4 THEORETICAL RESULTS

**Overview of Assumptions:** We provide an informal version of our assumptions here, with the formal ones stated in appendix A.1. We assume a bounded sensing range for the agent, along with bounded motions of all objects. Any practical environment has limits on the velocity of objects, thus these assumptions are fairly practical. For sample complexity results, we have basic assumptions over smoothness of $\pi_\theta(\cdot | \cdot)$, which are common in the literature (section A.1.4).

**Lemma 1** (Effective Resistance and Connectivity Relationship). *Under standard assumptions on the environment and graph construction (Assumptions 1-5 in the appendix), for any connected graph $\mathcal{G}_t$ containing the agent and goal, the temporal derivatives satisfy:*

$$\frac{d\mathcal{R}_{\text{eff}}(t)}{dt} \cdot \frac{d\kappa(\mathcal{G}_t)}{dt} \leq -C_1 \left| \frac{d\kappa(\mathcal{G}_t)}{dt} \right|^2$$

*for some constant $C_1 > 0$, where $\mathcal{R}_{\text{eff}}$ defined in eq 2 and $\kappa$ is the algebraic connectivity, ie the second smallest eigenvalue of the laplacian which we assume to be unique for simplicity.*

*This lemma provides the crucial insight for our intrinsic reward: decreasing effective resistance is mathematically linked to increasing graph connectivity. This allows us to use $\mathcal{R}_{eff}$ as a dense reward signal to guide the agent towards better configurations. While proven here, directly using $\kappa$ has two caveats though: (i) it provides only a coarse, environment-wide connectivity without focusing on the agent-goal topology, (ii) more importantly, we empirically observed that the variations in $\kappa$ are much noisier in contrast to $\mathcal{R}_{eff}$ as demonstrated in sections A.7, A.8.1.*

**Lemma 2** (Bounded Connectivity Change). *Under the same assumptions, the change in algebraic connectivity between consecutive timesteps is bounded:*

$$|\kappa(\mathcal{G}_{t+1}) - \kappa(\mathcal{G}_t)| \leq \delta_{\max}$$

*where $\delta_{\max}$ is a constant dependent on agent/object velocities and graph size bounds. We also show that $\kappa(\mathcal{G}_t)$ is Lipschitz continuous with respect to time in corollary 2*

*This result allows for relatively stable policy optimization.*

**Lemma 3** (Policy Updates). *The intrinsic reward $r_{int}(t)$ is positively correlated with the one-step change in algebraic connectivity, $\Delta\kappa(t) = \kappa(\mathcal{G}_{t+1}) - \kappa(\mathcal{G}_t)$.*

These supporting results culminate in our main theorem, which states that a policy optimized with our intrinsic reward is guaranteed to be robust.

**Theorem 1** (Connectivity Preservation and Robust Navigation). *Under our stated assumptions, a policy $\pi^*$ that maximizes the expected return with our effective resistance-based intrinsic reward is $(\epsilon, \delta, T)$-**robust**. Specifically:*

1. ***Connectivity Preservation**: The policy avoids actions that drastically reduce connectivity in expectation.*

2. ***Robust Navigation**: For a sufficiently large penalty/reward $\beta$ on goal visibility loss/gain, robustness (definition 5) of $\pi^*$ as the policy that reaches the goal with high probability while maintaining both goal visibility and graph connectivity, which we have also shown to hold very well empirically in sections A.8.1 and A.8.2*

**Corollary 1** (Practical Policy Design). *For effective implementation, the goal visibility penalty $\beta$ should be set to dominate the extrinsic reward, and the intrinsic reward weight $\alpha$ should be scaled relative to the environment's dynamics.*

All these results culminate in the hyperparameter choices detailed in appendix A.6 along with empirical justifications of the lemmas in subsequent appendix sections A.7, A.7.1, A.8, A.8.2

### 4.1 IMPROVED SAMPLE COMPLEXITY

**Lemma 4** (Variance Reduction via Intrinsic Reward). *The policy gradient for our combined objective ($r_{total}$) is an **almost unbiased estimator** for the extrinsic-only objective. Furthermore, $r_{int}(t)$ acts as a variance reduction baseline, which coupled with the sample complexity results from (Yuan et al., 2022) shows that policies trained with our formulation can achieve substantially faster convergence at*

$$U_{total} = \mathcal{O}(U(2 - 2\rho))$$

*where $U_{total}$ is the number gradient updates for our formulation and $U$ are gradient updates for vanilla PPO, $\rho$ is correlation coefficient between the $Q$ function at $a_t, s_t$ and negative of $-\alpha\mathcal{R}_{eff}(t)$*

*This result becomes particularly impactful in sparse reward settings where the extrinsic reward exhibits higher variance by default. The formal analysis is presented in Appendix A.1.*

## 5 EXPERIMENTS

**Environments Used:** We use Safety-Gymnasium (Ji et al., 2024) library as it provides sophisticated environments that serve as a great test suite.

- Navigation: The agent needs to navigate to the goal button. There are multiple obstacles that move arbitrarily in the environment thus hindering navigation. There are 3 difficulty levels - Level 0, Level 1 and Level 2 with increasing number of obstacles. We mark these environments as Navigation-Level-0, Navigation-Level-1 and Navigation-Level-2 respectively in the experiments hereafter.

- Building: This environment requires the agent to proficiently operate multiple machines within a construction site, while concurrently evading other robots and obstacles present in the area. Similar to the Navigation environment, we experiment on all three difficulty levels and report the numbers with same nomenclature.

- Fading: This environment requires the agent to reach the goal position, ensuring it steers clear of hazardous areas. However, the goal linearly disappears after 150 steps of the environment refresh, which provides an interesting and albeit unique challenge. Following a similar setup, we have three difficulty levels in this environment as well.

A visual representation of each of these environments is provided in section A.3

**Reward Sparsity**: In order to thoroughly assess the algorithms in a sparse reward setting, we utilize the *default reward* provided by the environments but at every $K$ steps (with $K = 25$ following works like (Memarian et al., 2021) for sparse reward settings). Thus, in the total reward formulation discussed in section 3.1 we have $r_{ext}(t) = r_{ext}(t) \times \mathbf{1}[t\%K = 0]$. This *default environment reward* is simply the difference of the agent to goal relative distance between timestep $t$ and $t + 1$, ie, $r_{ext} \geq 0$ if the agent moves closer to the goal (or stays at same distance) and negative otherwise. Another aspect that makes this reward setting more challenging is that this $r_{ext}(t)$ which is received at timestep $t$ is a local reward of improvement between state $t$ and $t+1$ and not between $t$ and $t-K$, thus providing a good testbed for high sparsity settings. We scale this reward by a factor of 10 to incorporate a noticeable $r_{ext}$ for the gradient estimations and policy updates.

**Baselines -** To perform an extensive evaluation of our method against the literature, we implement 6 baselines as follows. **PPO** (Schulman et al., 2017): which is the core algorithm that is being used to train most practical algorithms, especially the recent success of RL (Stiennon et al., 2022). **PPO + Ent** (Schulman et al., 2017): is the variation of PPO that explicitly incorporates the entropy loss over the action distribution obtained from the policy network, ie, $\pi_\theta(\cdot|s_t)$. **SRL (Surprise RL)** (Achiam & Sastry, 2017): is one of the initial works which proposed two different notions of surprise based intrinsic motivation by modeling the state transition probabilities explicitly via a separate network. These are added to the extrinsic reward $r_{ext}(t)$ with some weighting factor. The first one is the transition probability of $s_{t+1}$ given current state and action pair as $P_{\phi'}(s_{t+1}|a_t, s_t)$ (where the dynamics model $\phi'$ is parametrized as Gaussian MLP Regressor) whereas the second method learns a stream of the dynamics models $\phi'$ at different timesteps and then considers the intrinsic reward as $P_{\phi_t}(s_{t+1}|a_t, s_t) - P_{\phi_{t-z}}(s_{t+1}|a_t, s_t)$ for iteration difference $z$. We call the first version as **SRL-Std** for standard and the second version as **SRL-Diff**. **NGU** (Badia et al., 2020): proposed an episodic memory based intrinsic reward formulation. They construct an explicit explicit embedding model that is trained simultaneously, which however also leads to larger number of parameters across the pipeline in comparison to the other methods. **AIM** (Durugkar et al., 2021): constructs the intrinsic reward via the Wass-1 distance between a policy's state visitation distribution and a specific target distribution. We use the continuous state/action extension as discussed in their work. **MEGA** (Pitis et al., 2020): which is a goal conditioned RL algorithm that defines a distribution over the goals and enforces the exploration of the sparsely explored areas of the goal distribution. We implement the algorithm under the PPO setup to make it on-policy in order to perform a more pronounced comparison and leverage the policy improvements over the course of training. **GCPO** (Xudong et al., 2024): is a recent on-policy goal conditioned RL methodology which pretrains a goal conditioned policy via Behavioral Cloning and then perform an online self-curriculum style update to select goals from the MEGA based distribution fit and sampling from it to fine-tune further. Algorithm 1 of their work (Xudong et al., 2024) details the procedure. This method is on-policy by design.

**Evaluation Methodology:** We trained each of the models 5 times and then evaluated all the trained policies by rolling each of those 5 for 200 new episodes in the corresponding environments, thus providing a total of 1000 evaluation episodes in total and accounting for induced randomness due to initialization. For the main results in next section we compare the - (i) The percentage of successful episodes for each method where success is defined as the agent reaching the goal ; (ii) The median number of timesteps taken to reach the goal (the maximum number of timesteps per episode is

1000 as default setting in the Safety Gym library) along with the $25^{th}\%$ and $75^{th}\%$ marks; (iii) The median environment reward obtained by each agent across the episodes. This is computed by considering the average reward in each episode and then computing the median across the 1000 evaluation episodes (this implicitly takes the episode length into consideration and thus provides a default normalization). We provide the core results and convergence analysis here with the parameter selection discussion in A.6, evolution of $r_{int}$ during navigation in A.7, graph algorithm analysis in A.9, A.12, runtime analysis of the methods in A.11.

## 5.1 MAIN RESULTS

**Success rate:** Based on the results provided in figure 2, we notice a consistent outperformance of our method against the baselines, particularly in the more challenging settings. For example, at Navigation-level-2, our approach achieves a success rate of 55.5%, representing an absolute improvement of 16.8 percentage points over the strongest baseline (GCPO at 38.7%). At Building-level-2, our method reaches 88.4%, improving over GCPO by 32.7 points and more than double the performance of SRL-Std (38.8%). Even in moderately difficult tasks, such as Navigation-level-1 and Building-level-1, our method improves success rates to 99.6% and 99.2%, compared to the best baseline scores of 84.0% and 86.4%, respectively. On the easiest tasks (difficulty 0), performance across methods is already near ceiling, but our method remains competitive, matching or exceeding the baselines. A few noteworthy aspects are - (i) the fluctuation of the performance of PPO with entropy across the difficulty levels, where in some cases it substantially outperforms vanilla PPO, but much worse in Fading environment potentially due to over exploration while the goal vanishes in the meantime; (ii) GCPO first pretrains a policy using Behavioral Cloning and then fine tunes in an on-policy manner over trajectory distributions, whereas our method works from scratch which further substantiates our claims.

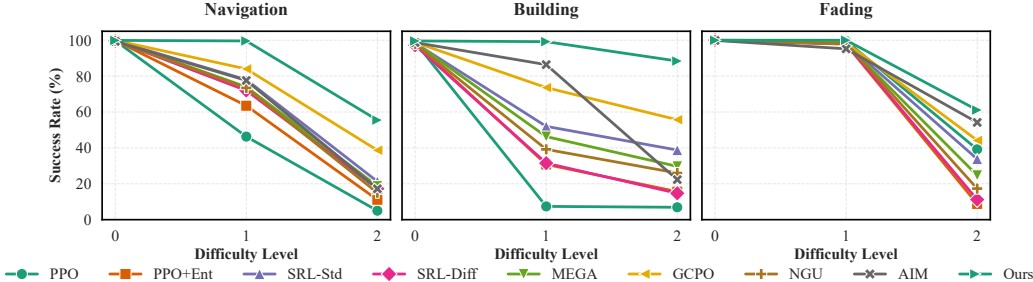

Figure 2: Success rate (% of successful episodes out of the 1000) during evaluation for all methods across Navigation (left), Building (center) and Fading (right) environments.

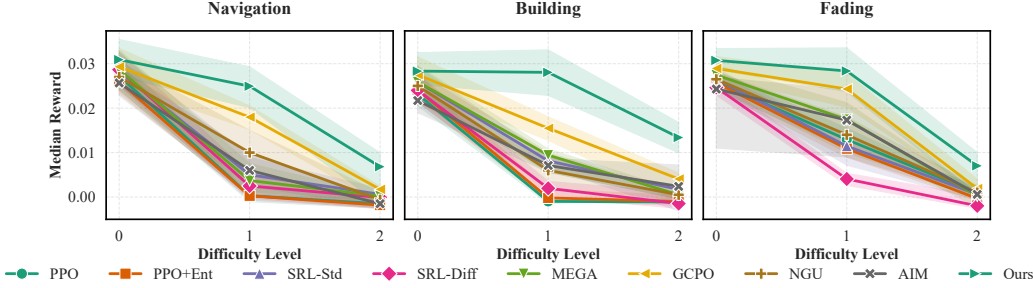

Figure 3: Median Reward during evaluation for all methods across Navigation (left), Building (center) and Fading (right) environments. The plots contain the $25^{th}\%$ and $75^{th}\%$ ticks.

**Median Normalized Reward:** The results are provided in figure 3. Across all the environments, our method achieves consistently higher normalized rewards, with particularly pronounced gains

in the more difficult settings. For example, at Navigation-level-2, our method reaches a median normalized reward of 0.0068, compared to 0.0017 for GCPO (a $4\times$ improvement) and this reward is near-zero or negative values for all other baselines. Notably, our $25^{th}$ percentile ($-0.0017$) is better than the median of PPO+Ent ($-0.0018$) and the PPO variants. Similarly, in Building-level-2, our approach yields 0.0134, which more than triples GCPO (0.0041) and surpasses the upper quartile of every baseline, while their medians often remain close to zero or negative. In Building-level-1, our method reaches 0.0281, nearly doubling GCPO (0.0155) and lying well above the $75^{th}$ percentile of all other methods, indicating a robust shift of the entire reward distribution. Even in moderately difficult environments such as Navigation-level-1 and Fading-level-1, our method maintains the highest scores (0.0249 and 0.0284, respectively), with our lower quartiles exceeding the medians of most baselines (e.g., our $25^{th}$ percentile 0.0201 at Navigation-level-1 surpasses the median of SRL-Std at 0.0050). On the easiest tasks (difficulty 0), all methods perform similarly, but our method consistently outperforms (e.g., 0.0309 at Navigation-level-0 vs 0.0293 for GCPO).

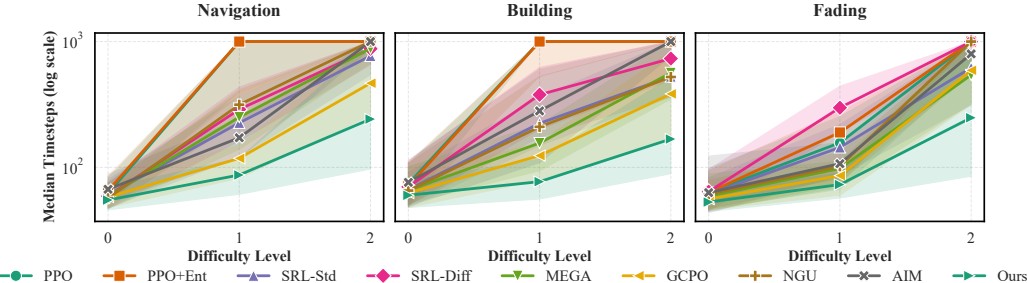

Figure 4: Median Number of Timesteps per episode during evaluation for all methods across Navigation (left), Building (center) and Fading (right) environments. The plots contain the $25^{th}\%$ and $75^{th}\%$ ticks

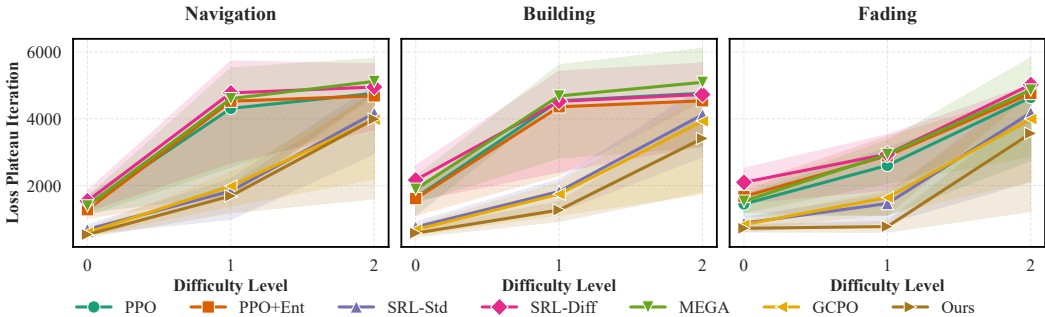

Figure 5: Comparison of the loss plateau iterations across the methods.

**Median Timesteps to reach the goal:** The results are provided in figure 4. While difficulty level 0 across the environments is relatively easier for all the methods, the levels 1 and 2 provide a vivid difference in the performance of the methods. For instance, at Navigation-level-2, our method requires a median of 242 steps, which almost half of GCPO (467) and far below Surprise RL variants at 769 and 881. Our median of 242 lies even below the $25^{th}$ percentile of GCPO which is at 258. Similarly, at Building-level-2, our method achieves 168 steps, whereas the next best (GCPO) requires 384, again placing our median earlier than the $25^{th}$ percentile of all other methods. In Building-level-1, our approach achieves a median of 77 steps, compared to GCPO's 124 and MEGA's 156, with our $75^{th}$ percentile (116.25) still lower than the median of the baselines. Even in moderately difficult settings, such as Navigation-level-1 and Fading-level-1, our method reduces the median steps to 87 and 73 respectively, with improvements of around 26% and 15% over GCPO (118 and 86), while remaining well below the lower quartiles of SRL variants (121 and 86). On the easiest levels (difficulty 0), where performance across methods is already near-optimal, our method still

achieves the lowest or near-lowest medians (e.g., 53 steps at Fading-level-0 vs. GCPO's 56). At the highest difficulty Fading-level-2, our method again demonstrates clear advantage, reducing the median to 248 compared to 543 (MEGA) to 1000 for other methods, with our $25^{th}$ percentile (85) substantially earlier than GCPO's 315. Furthermore, the median and at times the $25^{th}$ percentile timesteps hit the maximum limit of 1000 for the harder environment in some of the baselines, thus justifying the efficacy of our proposed intrinsic reward mechanism and its role in achieving the tasks. Furthermore, these significant performance improvements are achieved with high computational efficiency, incurring only a marginal runtime overhead (1.1x-1.25x vs. PPO) per episode training, as shown in our detailed runtime analysis in Appendix A.10, A.11. This is however completely counterbalanced by much faster training convergence of our method as seen in figure 5

## 5.2 Convergence Analysis

In order to empirically justify the claim in lemma 4, we perform two analysis here - (i) comparison of the *training* iterations at which the loss plateaus across the respective methods ; (ii) the correlation of our proposed $\mathcal{R}_{\text{eff}}$ (negative sign as discussed in the lemma) against the learned value function $V^{\pi}$ during *evaluation* phase of the same 1000 episodes (once the trained has completed, since it is much less practical to discuss about the learned value function otherwise). While our theoretical claims are offered for PG since it is harder to characterize PPO style algorithms, we still observe that our results hold well. We also point to remark 7 about the reason for computing these correlations against the value function.

The bar plots for plateau iterations are provided in figure 5. Uniformly across all settings, we observe a faster convergence than PPO, especially for easier environments where the correlations in figure 6 are higher (with negligible p-values). For the easier level-0 difficulty across the environments, GCPO and SRL-Std also exhibit faster convergence. However we note that GCPO leverages a pretrained policy during training which substantially increases its overall time. Furthermore, similar to the main results discussed earlier, PPO+Ent exhibits varying behavior against PPO in convergence as well. For Navigation and Building environments, we also note some of the baselines exhibiting very delayed convergence towards end

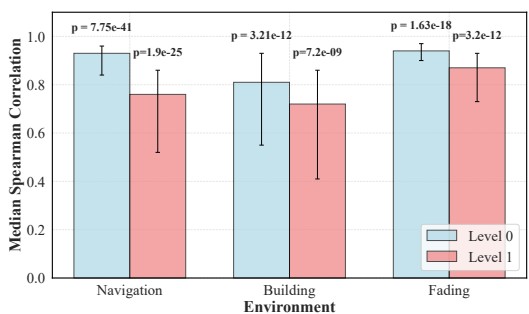

Figure 6: Spearman Correlation coefficients between the value function and $-\mathcal{R}_{\text{eff}}$.

of training iterations for levels 1 and 2. Quantitatively characterizing, for eg over Building-level-0 PPO has plateau at 1636 whereas ours at 576 thus the ratio of 0.35 whereas the corresponding correlation value of $2 - 2\rho = 0.38$, thus providing a very strong justification of lemma 4. We observe similar behavior for other environment and difficulty levels as well, albeit with some margin of error as discussed in remark 7 and accounting for the difficulty of approximation of the value and $Q$ functions (Moon et al., 2023; Engstrom et al., 2020).

## 6 Conclusion

We introduce a novel intrinsic reward formulation based on the notion of "Effective Resistance" from spectral graph theory. By providing extensive empirical results guided by the theoretical contributions, we show that this formulation works well across a diverse suite of tasks and only incurs a marginal increase in the runtime over baseline PPO but provides much faster convergence during training. We further conduct experiments to discuss the behavior of trained policies over the evolution of our intrinsic reward, goal visibility during navigation and analysis of key hyperparameters. We expect to drive the community's attention to graph based formulations, such as the one proposed in this work, as these provide a higher degree of interpretability as described earlier in figure 1 along with the performance improvements.

## 7 Reproducibility Statement

Adhering to the guidelines, we have provided all the relevant implementation details in section A.5. For the theoretical results, all the relevant assumptions, definitions and proofs are detailed in section A.1

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

# A APPENDIX

## A.1 DETAILS OF THEORETICAL ANALYSIS

This section provides the formal assumptions, definitions, remarks, and complete proofs for the theoretical results presented in Section 4.

### A.1.1 CORE ASSUMPTIONS

**Assumption 1** (Bounded Sensing Range). *The agent has a fixed sensing radius $R > 0$. Objects beyond distance $R$ from the agent are not included in the graph $\mathcal{G}_t$. The environment is bounded: agent and objects are confined to a domain $\Omega \subset \mathbb{R}^2$ with diameter $D < \infty$. This ensures finite graph sizes and well-defined computational procedures.*

**Assumption 2** (Consistent Graph Construction). *At each timestep $t$, the graph $\mathcal{G}_t = (V_t, E_t, W_t)$ is constructed deterministically relative to the agent using a fixed rule:*

- *All objects within radius $R$ are connected to the agent*

- *Edge weights represent the notion of vicinity/closeness, which we assume $\in (0, 1]$*

- *The goal is included if it is within $R$*

**Assumption 3** (Bounded Object Dynamics). *All objects (including the goal) move with bounded velocity: $\|\mathbf{v}_{obj}(t)\| \leq V_{\max}$. The agent's velocity is similarly bounded: $\|\mathbf{v}_{\mathcal{A}}(t)\| \leq v_{\max}$. This prevents arbitrary changes in graph structure. This assumption is realistic for any practical environment.*

**Assumption 4** (Connected Goal Path). *There exists at least one continuous path from any reachable agent position to the goal through free space when both are in the sensing range. This ensures the task is well-posed, as otherwise*

**Assumption 5** (Graph Size Bounds). *The number of nodes in $\mathcal{G}_t$ is bounded: $n_{\min} \leq |V_t| \leq n_{\max}$ for all $t$. This ensures computational tractability.*

**Assumption 6** (Goal Visibility Model). *The probability of maintaining goal visibility is a monotonically increasing function of the graph's current algebraic connectivity. We model this relationship as:*
$$\mathbb{E}[\mathbf{1}_{goal}(t+1)|\mathcal{G}_t] \geq 1 - \exp(-C_4\kappa(\mathcal{G}_t))$$
*for some constant $C_4 > 0$. This is justified as higher connectivity, which implies both a greater likelihood of maintaining proximity (via lower effective resistance) and a lower likelihood of goal path occlusion due to a greater number of redundant pathways. This exponential form is a standard modeling choice in network reliability and percolation theory (Colbourn, 1987; Bollobás, 2001).*

**Remark 1** (Justification for Assumption 6). *This assumption formalizes the intuitive link between the abstract graph connectivity and the tangible task objective of maintaining goal visibility. The justification rests on two physical components of visibility: proximity and occlusion, both of which are positively influenced by higher algebraic connectivity ($\kappa$).*

1. ***Proximity:*** *A policy that is incentivized to increase connectivity will naturally favor actions that maintain proximity to the goal. As established in Lemma 1, high algebraic connectivity ($\kappa$) corresponds to low effective resistance ($\mathcal{R}_{eff}$), and our intrinsic reward encourages the agent to minimize this resistance, thus reducing the "difficulty" of paths to the goal.*

2. ***Occlusion:*** *A more connected graph implies the existence of more redundant pathways between the agent and the goal. In the physical environment, this translates to having more alternative lines-of-sight, which reduces the probability of total occlusion by dynamic obstacles.*

*Given that the probabilities of maintaining proximity and avoiding occlusion are monotonically increasing functions of $\kappa$, their product (the probability of maintaining visibility) is also monotonically increasing. The exponential form $1 - \exp(-C_4\kappa)$ is a standard modeling choice for such phenomena because it respects probability bounds $[0, 1]$, captures the intuitive notion of diminishing returns, and is mathematically tractable. This functional form is well-established in related*

*fields such as network reliability analysis (Colbourn, 1987), wireless communication (Rappaport, 2002), and graph connectivity studies (Bollobás, 2001), validating its appropriateness for modeling connectivity-dependent events.*

### A.1.2 KEY DEFINITIONS

**Definition 1** (Graph Laplacian). *For graph $\mathcal{G}_t = (V_t, E_t, W_t)$ with $n$ nodes, the Laplacian matrix $L(t) \in \mathbb{R}^{n \times n}$ is defined as:*

$$L_{ij}(t) = \begin{cases} \sum_{k \neq i} w_{ik}(t) & \text{if } i = j \\ -w_{ij}(t) & \text{if } (i,j) \in E_t \\ 0 & \text{otherwise} \end{cases}$$

*The Laplacian is the core of graph theory (Zhang, 2011).*

**Definition 2** (Effective Resistance). *For nodes $i$ and $j$ in graph $\mathcal{G}_t$, the effective resistance is:*

$$\mathcal{R}_{\text{eff}}(i, j; \mathcal{G}_t) = (e_i - e_j)^T L^+(t)(e_i - e_j)$$

*where $e_k$ is the $k$-th standard basis vector. This measures the "difficulty" of information flow between nodes $i$ and $j$.*

**Definition 3** (Algebraic Connectivity). *The algebraic connectivity of graph $\mathcal{G}_t$ is:*

$$\kappa(\mathcal{G}_t) = \lambda_2(L(t))$$

*where $\lambda_2$ is the second-smallest eigenvalue of the Laplacian. Higher values indicate better connectivity. For disconnected graphs, $\kappa(\mathcal{G}_t) = 0$.*

**Definition 4** (Edge weights $w_{i,j}$). *The edge weights in $W_t$ are a function of the state/observation vector $s_t$, ie,*

$$w_{i,j}(t) = h_{i,j}(s_t)$$

*described via algorithm 1. Here, we loosely define the function via indexing $(i, j)$, but based on the aforementioned assumptions, all such $h_{i,j}$*

1. *are continuous*

2. *Lipschitz w.r.t to the input $s_t$ thus*

$$|h_{i,j}(s_t) - h_{i,j}(s_{t'})| \leq \hat{\mathcal{L}}|s_t - s_{t'}|$$

*and since the change is the state vectors is also bounded, we have that*

$$|h_{i,j}(s_t) - h_{i,j}(s_{t'})| \leq \bar{\mathcal{L}}|t - t'|$$

*where $\hat{\mathcal{L}}$ and $\bar{\mathcal{L}}$ are the respective Lipschitz constants*

3. *bounded between $(0, 1]$ (notice 0 exclusive as discussed previously)*

**Definition 5** (Robust Navigation Policy). *A policy $\pi$ is $(\epsilon, \delta, T)$-robust if:*

1. *It reaches the goal with probability $\geq 1 - \epsilon$ within $T$ steps*

2. *It maintains goal visibility: $\mathbb{E}_{\tau \sim \pi}[\sum_{t=0}^{T} \mathbf{1}_{goal}(t)] \geq (1 - \delta)T$*

3. *It preserves connectivity: $\mathbb{E}_{\tau \sim \pi}[\kappa(\mathcal{G}_t)] \geq \delta$ for all $t$*

*This definition captures both task success and structural stability.*

### A.1.3 PROOFS FOR MAIN RESULTS

Hereafter, we denote $\mathcal{R}_{\text{eff}}$ as the Effective Resistance between agent and goal node, ie, $\mathcal{R}_{\text{eff}}(t) = \mathcal{R}_{\text{eff}}(\mathcal{A}, g; \mathcal{G}_t)$

**Lemma 1**: Under Assumptions 1-5, the temporal derivatives of effective resistance and connectivity satisfy:

$$\frac{d\mathcal{R}_{\text{eff}}(t)}{dt} \cdot \frac{d\kappa(\mathcal{G}_t)}{dt} \leq -C_1 |\frac{d\kappa(\mathcal{G}_t)}{dt}|^2$$

.

*Proof.* The proof establishes the inverse relationship between the rates of change of effective resistance and algebraic connectivity by analyzing the time derivative of the spectral decomposition of $\mathcal{R}_{\text{eff}}$.

**Step 1: Differentiability Framework.** As established in the lemma statement, we consider a continuously evolving graph $\mathcal{G}_t$. As the positions of all nodes are differentiable functions of time, making the edge weights $w_{ij}(t)$ and consequently the Laplacian matrix $L(t)$ differentiable. For generic graph perturbations, the second eigenvalue $\kappa(t) = \lambda_2(t)$ is simple (has multiplicity 1), which ensures that $\kappa(t)$ and its corresponding eigenvector $v_2(t)$ are differentiable functions of time.

**Step 2: Spectral Decomposition.** Our starting point is the spectral representation of the effective resistance between the agent node $\mathcal{A}$ and the goal node $g$:

$$\mathcal{R}_{\text{eff}} = \sum_{i=2}^{n} \frac{1}{\lambda_i(t)} \langle e_{\mathcal{A}} - e_g, v_i(t) \rangle^2$$

where $\lambda_i(t)$ and $v_i(t)$ are the eigenvalues and orthonormal eigenvectors of $L(t)$, respectively. From this definition, $\mathcal{R}_{\text{eff}}$ is also differentiable wrt time.

**Step 3: Time Derivative Calculation.** We now take the total derivative of $\mathcal{R}_{\text{eff}}$ with respect to time $t$. Applying the chain rule to each term in the summation:

$$\frac{d\mathcal{R}_{\text{eff}}}{dt} = \sum_{i=2}^{n} \left[ \frac{d}{dt} \left( \frac{1}{\lambda_i(t)} \right) \cdot \langle e_{\mathcal{A}} - e_g, v_i(t) \rangle^2 + \frac{1}{\lambda_i(t)} \cdot \frac{d}{dt} \left( \langle e_{\mathcal{A}} - e_g, v_i(t) \rangle^2 \right) \right]$$

Let's analyze the derivative of each part:

- The derivative of the eigenvalue term is: $\frac{d}{dt} \left( \frac{1}{\lambda_i(t)} \right) = -\frac{1}{\lambda_i(t)^2} \frac{d\lambda_i(t)}{dt}$.

- The derivative of the eigenvector term involves the derivative of the eigenvectors, $\frac{dv_i}{dt}$.

Substituting the first part back, we get:

$$\frac{d\mathcal{R}_{\text{eff}}}{dt} = \sum_{i=2}^{n} \left[ -\frac{1}{\lambda_i(t)^2} \frac{d\lambda_i(t)}{dt} \langle e_{\mathcal{A}} - e_g, v_i(t) \rangle^2 \right] + \text{terms involving } \frac{dv_i}{dt}$$

For graph evolutions in which the dominant effect on the change in resistance comes from the change in the eigenvalues, particularly the smallest non-zero one, $\lambda_2(t) = \kappa(t)$, the change in eigenvectors typically has a smaller impact. Thus, we can approximate the relationship by focusing on the most significant term in the sum, which is the term for $i = 2$:

$$\frac{d\mathcal{R}_{\text{eff}}}{dt} = -\frac{1}{\lambda_2(t)^2} \frac{d\lambda_2(t)}{dt} \langle e_{\mathcal{A}} - e_g, v_2(t) \rangle^2 + \mathcal{O}(\epsilon_0) = -\frac{1}{\kappa(t)^2} \frac{d\kappa(t)}{dt} \langle e_{\mathcal{A}} - e_g, v_2(t) \rangle^2 + \mathcal{O}(\epsilon_0)$$

for $\dfrac{\mathcal{O}(\epsilon_0)}{-\frac{1}{\kappa(t)^2} \frac{d\kappa(t)}{dt} \langle e_{\mathcal{A}} - e_g, v_2(t) \rangle^2} \to 0$

**Step 4: Sign Analysis.** We now analyze the terms in the relationship derived above to establish the final inequality. Let's define a term $C(t) = \frac{1}{\kappa(t)^2} \langle e_{\mathcal{A}} - e_g, v_2(t) \rangle^2$.

- For a connected graph, the algebraic connectivity $\kappa(t) = \lambda_2(t)$ is strictly positive. Therefore, $\frac{1}{\kappa(t)^2} > 0$.

- The term $\langle e_{\mathcal{A}} - e_g, v_2(t) \rangle^2$ represents the squared projection of the vector $(e_{\mathcal{A}} - e_g)$ onto the Fiedler vector (Fiedler, 1973) $v_2(t)$. The vector $(e_{\mathcal{A}} - e_g)$ is orthogonal to the first eigenvector $v_1(t)$ (the constant vector). For any non-pathological connected graph, $(e_{\mathcal{A}} - e_g)$ is not orthogonal to the Fiedler vector $v_2(t)$, ensuring that $\langle e_{\mathcal{A}} - e_g, v_2(t) \rangle^2 > 0$ with high probability.

Since both components of $C(t)$ are positive, $C(t) > 0$. Our relationship is thus:

$$\frac{d\mathcal{R}_{\text{eff}}}{dt} \approx -C(t)\frac{d\kappa(t)}{dt}$$

To arrive at the form in the lemma statement, we multiply both sides by $\frac{d\kappa(t)}{dt}$:

$$\left(\frac{d\mathcal{R}_{\text{eff}}}{dt}\right)\left(\frac{d\kappa(t)}{dt}\right) \approx -C(t)\left(\frac{d\kappa(t)}{dt}\right)^2 = -C(t)\left|\frac{d\kappa(t)}{dt}\right|^2$$

Since $C(t)$ is strictly positive and bounded below by some constant $C_1$ over the compact state space over which we have the possible graph configurations, we obtain:

$$\frac{d\mathcal{R}_{\text{eff}}}{dt}\frac{d\kappa(t)}{dt} \leq -C_1\left|\frac{d\kappa(t)}{dt}\right|^2$$

This establishes the inverse relationship: when connectivity increases ($\frac{d\kappa}{dt} > 0$), effective resistance decreases ($\frac{d\mathcal{R}_{\text{eff}}}{dt} < 0$), and vice-versa. □

**Remark 2.** *Lemma 1 establishes that higher connectivity directly reduces effective resistance, making the goal more accessible. This is intuitive: more connections provide alternative paths, reducing the "electrical resistance" between agent and goal.*

**Lemma 2** Under Assumptions 1-5, for any action $a_t$:

$$|\kappa(\mathcal{G}_{t+1}) - \kappa(\mathcal{G}_t)| \leq \delta_{\max}$$

where $\delta_{\max} = C_2 \cdot \Delta w_{\max} + k$ for constants $C_2, C_3 > 0$, and $k$ is the maximum number of nodes that can be added/removed per timestep (we discuss its practicality in section A.6).

*Proof.* The change in the graph structure from $\mathcal{G}_t$ to $\mathcal{G}_{t+1}$ is a result of two distinct processes: (1) discrete changes in the graph's topology as nodes and edges are added or removed, and (2) continuous changes in edge weights due to the movement of the agent and objects. We will bound the impact of each process on the algebraic connectivity $\kappa(t)$ and then combine them.

**Case 1: Bounding the Effect of Topological Changes (Node Addition/Removal).** This component bounds the change in connectivity when the set of vertices $V_t$ changes.

- **Bounding the Number of Node Changes ($k$):** Nodes are added or removed when objects cross the agent's sensing radius $R$. By Assumption 3, the agent's velocity is bounded by $v_{\max}$ and object velocities are bounded by $V_{\max}$. The maximum relative speed between the agent and any object is thus $v_{\max} + V_{\max}$. In a time interval $\Delta t$, the number of objects that can cross agent's radial boundary is finite and bounded, as it depends on this maximum speed and the density of objects in the environment (which is finite by Assumption 5). This implicitly bounds the value of $k$ by algorithm 1.

- **Bounding the Connectivity Change:** Standard results in spectral graph theory provide bounds on the change in $\kappa$ when $k$ nodes are removed or added. A simple bound is $|\Delta\kappa| \leq k \cdot w_{\max}$, where $w_{\max}$ is the maximum possible edge weight, with $0 < w_{i,j} \leq 1$ as discussed earlier.

Combining these, the maximum change in connectivity due to topological changes, which we denote $|\Delta\kappa|_{\text{nodes}}$, is bounded by a constant factor of $k$:

$$|\Delta\kappa|_{\text{nodes}} \leq k$$

**Case 2: Bounding the Effect of Edge Weight Perturbation.** This component bounds the change in connectivity when the graph topology is fixed but edge weights change due to movement.

- **Bounding Weight Changes ($\Delta w_{\max}$):** As previously discussed, since $0 < w_{i,j} \leq 1$, thus $0 \leq \Delta w_{\max} < 1$

- **Applying Matrix Perturbation Theory:** By Weyl's inequality, the change in any eigenvalue of a Hermitian matrix is bounded by the spectral norm of the matrix perturbation. Applying this to the Laplacian matrix $L(t)$:

$$|\kappa(\mathcal{G}_{t+1}) - \kappa(\mathcal{G}_t)| = |\lambda_2(L_{t+1}) - \lambda_2(L_t)| \leq \|L_{t+1} - L_t\|_2 = \|\Delta L\|_2$$

The spectral norm $\|\Delta L\|_2$ is itself bounded by a function of $\Delta w_{\max}$ and the graph's maximum degree (which is bounded by $n_{\max}$ under Assumption 5). This relationship can be expressed as $\|\Delta L\|_2 \leq C_2 \cdot \Delta w_{\max}$ for some constant $C_2$ which depends on the environment dynamics.

Combining these, the maximum change in connectivity due to edge weight perturbations, denoted $|\Delta\kappa|_{\text{weights}}$, is:

$$|\Delta\kappa|_{\text{weights}} \leq C_2 \cdot \Delta w_{\max} \leq C_2$$

**Combined Effect.** To find the total bound, we use the triangle inequality. Let $\mathcal{G}'_t$ be a hypothetical graph with the same nodes as $\mathcal{G}_t$ but with the edge weights of $\mathcal{G}_{t+1}$. The total change can be decomposed as:

$$|\kappa(\mathcal{G}_{t+1}) - \kappa(\mathcal{G}_t)| = |\kappa(\mathcal{G}_{t+1}) - \kappa(\mathcal{G}'_t) + \kappa(\mathcal{G}'_t) - \kappa(\mathcal{G}_t)|$$

$$\leq \underbrace{|\kappa(\mathcal{G}_{t+1}) - \kappa(\mathcal{G}'_t)|}_{\text{Effect of node changes}} + \underbrace{|\kappa(\mathcal{G}'_t) - \kappa(\mathcal{G}_t)|}_{\text{Effect of weight changes}}$$

The first term is bounded by $|\Delta\kappa|_{\text{nodes}}$ and the second term is bounded by $|\Delta\kappa|_{\text{weights}}$. Therefore, the total change is bounded by the sum of the individual bounds:

$$|\kappa(\mathcal{G}_{t+1}) - \kappa(\mathcal{G}_t)| \leq |\Delta\kappa|_{\text{weights}} + |\Delta\kappa|_{\text{nodes}} \leq C_2 \cdot \Delta w_{\max} + k$$

This sum defines the maximum possible change in connectivity in a single time step, $\delta_{\max}$. $\qquad\square$

**Corollary 2** (Continuity and Lipschitzness of $\kappa$ almost surely, a.s.)**.** *The connectivity function $\kappa(\mathcal{G}_t)$ is continuous and Lipschitz continuous with respect to time $t$*

*Proof.* This corollary is a direct consequence of the bounded change in connectivity established in Lemma 2. We prove each property separately.

**Part 1: Proof of Continuity**

To prove continuity at a time $t$, we must show that $\lim_{\Delta t \to 0} |\kappa(\mathcal{G}_{t+\Delta t}) - \kappa(\mathcal{G}_t)| = 0$.

From the result of Lemma 2, for any time interval $\Delta t > 0$, the change in connectivity is bounded by:

$$|\kappa(\mathcal{G}_{t+\Delta t}) - \kappa(\mathcal{G}_t)| \leq \delta_{\max}$$

Substituting the expression for $\delta_{\max}$:

$$|\kappa(\mathcal{G}_{t+\Delta t}) - \kappa(\mathcal{G}_t)| \leq C_2 \cdot \Delta w_{\max}(\Delta t) + k(\Delta t)$$

where we define $\Delta w_{\max}(\Delta t)$ as the change in the maximum edge weight difference and $k(\Delta t)$ is the number of nodes that can enter or leave the sensing radius within the time interval $\Delta t$.
We discuss each of these separately:

1. As defined earlier in definition 4, we have $\Delta w_{\max}(\Delta t) \leq \bar{\mathcal{L}}\Delta t$

2. For $k(\Delta t)$ we note that as the time interval shrinks ($\Delta t \to 0$), the maximum distance any object can travel also shrinks. While there exists some cases when the $k$ can change somewhat arbitrarily even as $\Delta t \to 0$ (such as when all objects are moving radially inwards towards the agent and abruptly cross cross within the sensing radius $R$), considering all possibilities the probability of such events $P(\mathcal{E}) = 0$ , where $\mathcal{E} : \{k(\Delta t) \gg n_{\min}\}$. Loosely speaking, such events have measure 0 and **never occur** almost surely, a.s and thus we have almost surely that $\lim_{\Delta t \to 0} k(\Delta t) = 0$

Now, taking the limit of our inequality as $\Delta t \to 0$:

$$\lim_{\Delta t \to 0} |\kappa(\mathcal{G}_{t+\Delta t}) - \kappa(\mathcal{G}_t)| \leq \lim_{\Delta t \to 0} [C_2 \Delta w_{\max}(\Delta t) + k(\Delta t)] = 0$$

which establishes the continuity for the function $\kappa(t)$ almost surely.

**Part 2: Proof of Lipschitz Property**

To prove Lipschitz continuity, we must show that there exists a constant $\mathcal{L}_\kappa > 0$ such that for any two times $t_1$ and $t_2$:

$$|\kappa(\mathcal{G}_{t_2}) - \kappa(\mathcal{G}_{t_1})| \leq \mathcal{L}_\kappa |t_2 - t_1|$$

First, we establish a bound on the magnitude of the time derivative of connectivity, $|\frac{d\kappa}{dt}|$. From the inequality in Part 1, we can divide by $\Delta t$:

$$\frac{|\kappa(\mathcal{G}_{t+\Delta t}) - \kappa(\mathcal{G}_t)|}{\Delta t} \leq C_2 \cdot \bar{\mathcal{L}} + \frac{k(\Delta t)}{\Delta t}$$

Taking the limit as $\Delta t \to 0$, the left side becomes the definition of the magnitude of the derivative. The term $\frac{k(\Delta t)}{\Delta t}$ represents the instantaneous rate of nodes crossing the sensing boundary, which *almost surely* (as discussed above) is a finite value bounded by the environment's dynamics. Let's call the maximum possible rate $k_{\text{rate}}$.

$$\left| \frac{d\kappa}{dt} \right| \leq C_2 \cdot \bar{\mathcal{L}} + k_{\text{rate}}$$

This shows that the derivative of the connectivity function is bounded. Let's define this upper bound as the Lipschitz constant, $L_\kappa$:

$$\mathcal{L}_\kappa := C_2 \cdot \bar{\mathcal{L}} + k_{\text{rate}}$$

Now, using the Fundamental Theorem of Calculus for any $t_1, t_2$ (assuming $t_1 < t_2$ without loss of generality):

$$\kappa(t_2) - \kappa(t_1) = \int_{t_1}^{t_2} \frac{d\kappa(s)}{ds} ds$$

Taking the absolute value of both sides and applying the triangle inequality for integrals ($|\int f ds| \leq \int |f| ds$):

$$|\kappa(t_2) - \kappa(t_1)| = \left| \int_{t_1}^{t_2} \frac{d\kappa(s)}{ds} ds \right| \leq \int_{t_1}^{t_2} \left| \frac{d\kappa(s)}{ds} \right| ds$$

We can now substitute our bound for the derivative's magnitude:

$$|\kappa(t_2) - \kappa(t_1)| \leq \int_{t_1}^{t_2} \mathcal{L}_\kappa ds$$

Evaluating the simple integral gives:

$$|\kappa(t_2) - \kappa(t_1)| \leq \mathcal{L}_\kappa |t_2 - t_1|$$

This result can be written for any $t_1, t_2$ as $|\kappa(t_2) - \kappa(t_1)| \leq L_\kappa |t_2 - t_1|$, which proves that $\kappa(t)$ is almost surely Lipschitz with constant $\mathcal{L}_\kappa$. $\qquad \square$

**Remark 3.** *Lemma 2 ensures that connectivity cannot change arbitrarily fast, providing stability for the learning process. The bound increases with agent/object speeds and decreases with minimum graph size, which is intuitive.*

**Lemma 3** (Policy Updates): The intrinsic reward $r_{\text{int}}(t)$ is positively correlated with the one-step change in algebraic connectivity, $\Delta\kappa(t) = \kappa(\mathcal{G}_{t+1}) - \kappa(\mathcal{G}_t)$.

*Proof.* The proof relies on showing that actions which increase connectivity receive higher intrinsic rewards in expectation. We analyze the immediate reward $r_{\text{int}}(t)$ based on the outcome of an action.

1. **Case 1: Goal Visible ($\mathbf{1}_{\text{goal}}(t) = 1$, $\mathbf{1}_{\text{goal}}(t+1) = 1$):** The reward is $r_{\text{int}}(t) = -\Delta\mathcal{R}_{\text{eff}}(t)$. From Lemma 1, an increase in connectivity ($\Delta\kappa > 0$) implies a decrease in effective resistance ($\Delta\mathcal{R}_{\text{eff}} < 0$). Thus, in this case, $r_{\text{int}} > 0$.

2. **Case 2: Goal Lost ($\mathbf{1_{goal}}(t) = 1, \mathbf{1_{goal}}(t+1) = 0$):** The reward is a large penalty $r_{int}(t) = -\beta$. Losing the goal node constitutes a major degradation of the graph structure, causing a significant drop in connectivity ($\Delta\kappa \ll 0$).

3. **Case 3: Goal Recovery ($\mathbf{1_{goal}}(t) = 0, \mathbf{1_{goal}}(t+1) = 1$):** The reward is a large bonus $r_{int}(t) = +\beta$. Recovering the goal node is a major structural improvement, causing a significant increase in connectivity ($\Delta\kappa \gg 0$).

In all cases, there is a strong positive correlation between the sign of $r_{int}(t)$ and the sign of $\Delta\kappa(t)$. Therefore, actions that are expected to increase connectivity will yield a higher expected intrinsic reward. $\qquad\square$

**Remark 4.** *Lemma 3 shows that our intrinsic reward is well-posed. Since policy gradient methods update policies to increase the likelihood of actions that lead to higher rewards, this positive correlation ensures the agent will learn to favor actions that preserve or enhance the graph's structural connectivity.*

**Theorem 1** Under Assumptions 1-6, a policy $\pi^*$ that maximizes the expected return with our effective resistance-based intrinsic reward is $(\epsilon, \delta, T)$-robust.

*Proof.* **Part 1: Connectivity Preservation**

1. **Optimality Condition:** Say the policy $\pi^*$ maximizes $J(\pi) = \mathbb{E}_{\zeta\sim\pi}[\sum \gamma^t(r_{ext}(t) + \alpha r_{int}(t))]$. At a local maximum, the policy gradient is zero: $\nabla_\theta J(\pi^*) = 0$. This implies a balance between the gradients of the different reward components: $\alpha\nabla_\theta J_{int}(\pi^*) = -\nabla_\theta J_{ext}(\pi^*)$.

2. **Expected Change in Connectivity:** From Lemma 3 we know that increasing intrinsic rewards (by virtue of policy updates that maximize the discounted reward) further leads to trajectories that increase connectivity $\kappa$. It will avoid actions that needlessly decrease connectivity, especially if an alternative exists that is neutral or beneficial to the extrinsic reward.

3. **Bounding the Expected Change:** The worst possible single-step decrease in connectivity is physically bounded by $\delta_{max}$, as shown in Lemma 2. Since the policy $\pi^*$ is optimized to maximize reward, and actions leading to lower $\kappa$ are penalized by $r_{int}$, the policy will learn to avoid these worst-case outcomes. Therefore, the expected change in connectivity under $\pi^*$ must be no worse than this physical bound: $\mathbb{E}_{\tau\sim\pi^*}[\Delta\kappa] \geq -\delta_{max}$. Substituting this gives the desired result: $\mathbb{E}_{\tau\sim\pi^*}[\kappa(\mathcal{G}_{t+1})] \geq \kappa(\mathcal{G}_t) - \delta_{max}$

**Part 2: Robust Navigation** We demonstrate that $\pi^*$ satisfies the three conditions of Definition 5.

1. **Goal Visibility Maintenance:** By choosing $\beta$ to be sufficiently large ( Corollary 1), the penalty for goal loss, $-\beta$, dominates all other single-step rewards or costs. An optimal policy $\pi^*$ will be strongly driven to select actions that maintain goal visibility to avoid the catastrophic $-\beta$ penalty. This directly ensures that the expected time the goal is visible is high, satisfying condition 2: $\mathbb{E}[\sum \mathbf{1_{goal}}(t)] \geq (1-\delta)T$.

2. **Connectivity Preservation:** This follows from Part 1 of this theorem and the result above. Since the agent maintains goal visibility most of the time, its behavior is primarily governed by the intrinsic reward $r_{int} = -\Delta\mathcal{R}_{eff}$. As established, optimizing this reward preserves connectivity. Thus, the expected connectivity under $\pi^*$ will remain above some positive threshold $\delta_\kappa > 0$, satisfying condition 3 above.

3. **Goal Reaching Probability:** With goal visibility maintained (from point 1) and connectivity preserved (from point 2), the agent has a consistent view of the goal and receives a dense, informative reward signal ($r_{int}$) that guides it towards more "accessible" configurations. This effectively transforms the sparse reward problem into one with a continuous guidance signal. Standard RL convergence results apply, showing that an agent in this

setting will learn a policy that reaches the goal with high probability, $\geq 1 - \epsilon$, satisfying condition 1.

Since $\pi^*$ satisfies all three criteria, it is, by definition, an $(\epsilon, \delta, T)$-robust policy. $\qquad\square$

**Remark 5.** *Theorem 1 establishes that our effective resistance-based intrinsic motivation naturally leads to robust navigation policies. The key insight is that optimizing for goal accessibility (via effective resistance) inherently promotes graph connectivity, which in turn supports reliable goal reaching and visibility maintenance.*

**Corollary 1** For practical implementation, choose $\beta \geq C \cdot \max_{s,a} |r_{\text{ext}}(s,a)|$ and $\alpha \geq C' \cdot \frac{\max_{s,a} |r_{\text{ext}}(s,a)|}{\delta_{\max} C_1}$.

*Proof.* This corollary provides design principles derived directly from the theoretical results.

- **Choice of $\beta$:** The role of the $\pm\beta$ reward is to make goal visibility transitions the most salient events for the learning agent. To guarantee that the agent prioritizes maintaining or regaining goal visibility over any other immediate reward, the magnitude of $\beta$ must exceed the maximum possible value of any other reward or cost achievable in a single step. Setting $\beta \geq C \cdot \max_{s,a} |r_{\text{ext}}(s,a)|$ for a safety factor $C > 1$ formally enforces this dominance.

- **Choice of $\alpha$:** The role of $\alpha$ is to scale the intrinsic reward's influence on the policy updates. This influence must be strong enough to promote exploration based on connectivity, counteracting small extrinsic rewards that might otherwise lead the agent to degrade its connectivity. The term $\delta_{\max}$ from Lemma 2 quantifies the maximum possible one-step degradation of connectivity, while $C_1$ from Lemma 1 quantifies the sensitivity of the reward signal to connectivity changes. Combining these with the extrinsic reward value, the ratio $\frac{\max_{s,a} |r_{\text{ext}}(s,a)|}{\delta_{\max} C_1}$ thus provides a natural scale for how much "work" the intrinsic reward must do. Setting $\alpha$ proportional to this ratio ensures the connectivity-preserving signal is appropriately weighted against the environment's dynamics, with the safety factor $C'$ providing an additional buffer.

$\qquad\square$

**Remark 6.** *Theorem 1 establishes that our effective resistance-based intrinsic motivation naturally leads to robust navigation policies. The key insight is that optimizing for goal accessibility (via effective resistance) inherently promotes graph connectivity, which in turn supports reliable goal reaching and visibility maintenance.*

### A.1.4 DETAILS OF CONVERGENCE AND SAMPLE COMPLEXITY ANALYSIS

Here, we provide the background and proof for the sample complexity result in Section 4.1. We leverage the general sample complexity framework for policy gradient methods established by (Yuan et al., 2022).

**Definition 6** (Policy Gradient Estimator)**.** *Let*

$$J_{ext}(\theta) = \mathbb{E}_{\zeta \sim \pi_\theta}\Big[\sum_{t=0}^{\infty} \gamma^t r_{ext}(t)\Big]$$

*be the expected extrinsic return and*

$$J_{T,ext}(\theta) = \mathbb{E}_{\zeta \sim \pi_\theta}\Big[\sum_{t=0}^{T} \gamma^t r_{ext}(t)\Big]$$

*be the truncated version of it*
*Additionally, let the GPOMDP gradient estimator Baxter & Bartlett (2001) for the finite horizon $T$ and with $m$ trajectories be:*

$$\hat{\nabla}_m J_{T,ext}(\theta) = \frac{1}{m} \sum_{i=1}^{m} \sum_{t=0}^{T-1} \left(\sum_{k=0}^{t} \nabla_\theta \log \pi_\theta(a_k^{(i)}|s_k^{(i)})\right) \cdot \gamma^t r_{ext}^{(i)}(t)$$

*Similarly, for our method, the estimator for the total reward $r_{total} = r_{ext} + \alpha r_{int}$ can be denoted as $\hat{\nabla}_m J_{total}(\theta)$.*

**Assumption 7** (Smoothness and Truncation, assumption 3.1 and 3.2 (Yuan et al., 2022)). *The objective function $J_{ext}(\theta)$ is Lipschitz-smooth with the constant $G > 0$. The truncation error of the finite-horizon objective $J_{T,ext}(\theta)$ w.r.t the original objective $J_{ext}(\theta)$ is bounded by constants $D, D' > 0$.*

**Assumption 8** (ABC Condition (Khaled & Richtárik, 2020)). *Let $J^* = \sup_\theta J_{ext}(\theta)$. There exist non-negative constants $A, B, C$ such that the policy gradient estimator $\hat{\nabla}_m J(\theta)$ satisfies:*

$$\mathbb{E}[\|\hat{\nabla}_m J_{T,ext}(\theta)\|^2] \le 2A(J^* - J_{ext}(\theta)) + B\|\nabla J_{T,ext}(\theta)\|^2 + C$$

Under these conditions, the sample complexity to find an $\epsilon$-approximate stationary point is derived by (Yuan et al., 2022) as follows:

**Theorem 2** (Sample Complexity of Policy Gradient, Thm 3.4 in Yuan et al. (2022)). *Under Assumptions 7 and 8, to find a point $\theta$ such that $\mathbb{E}[\|\nabla J(\theta)\|^2] \le \epsilon$, the number of iterations $Q$ required is bounded by:*

$$\tag{3}$$

$$U \ge \frac{12\delta_0 G}{\epsilon^2} \cdot \max\left\{B, \frac{12\delta_0 A}{\epsilon^2}, \frac{2C}{\epsilon^2}\right\}$$

*where $\delta_0 = J^* - J(\theta_0)$ is the initial suboptimality gap.*

**Proof of variance reduction in lemma 4 with sample complexity bounds using $r_{int}$**

**Lemma 4** For the intrinsic reward be $r_{int}(t) = \mathcal{R}_{\text{eff}}(t) - \mathcal{R}_{\text{eff}}(t+1)$. The policy gradient estimator for the total reward satisfies:

1. **Almost Unbiased Estimator:** $\mathbb{E}[\hat{\nabla}_m J_{T,\text{total}}(\theta)] \approx \nabla J_{T,ext}(\theta)$.

2. **Variance Reduction:** $C_{total} \approx C_{ext}(1 - \rho^2)$, where $\rho = \text{Corr}(Q_{ext}(s_t, a_t), -\mathcal{R}_{\text{eff}}(s_t))$.

*Proof.* **Part 1: Almost Unbiased Gradient Estimator:** The sum of intrinsic rewards from time $t$ to a finite horizon $T$ forms a telescoping series:

$$\sum_{k=t}^{T-1} r_{\text{int}}(k) = \sum_{k=t}^{T-1} \gamma^{k-t}(\mathcal{R}_{\text{eff}}(k) - \mathcal{R}_{\text{eff}}(k+1)) \tag{4}$$

$$= \mathcal{R}_{\text{eff}}(t) + \sum_{k=t+1}^{T-1} \gamma^{k-t-1} \times (\gamma - 1)\mathcal{R}_{\text{eff}}(s_k) - \gamma^{T-t-1}\mathcal{R}_{\text{eff}}(s_T) \tag{5}$$

As the value of $\gamma \approx 1$ (0.99 used in our experiments and commonly across the literature, table 2), we note that all the terms but $\mathcal{R}_{\text{eff}}(t)$ become negligible in comparison, ie $\sum_{k=t+1}^{T-1} \gamma^{k-t-1} \times (\gamma - 1)\mathcal{R}_{\text{eff}}(s_k) - \gamma^{T-t-1}\mathcal{R}_{\text{eff}}(s_T) \to 0$. Thus the total reward-to-go admits the form $Q_{\text{total}}(s_t, a_t) \approx Q_{\text{ext}}(s_t, a_t) + \alpha\mathcal{R}_{\text{eff}}(t)$. Since the second term depends only on the current state, we can consider $b(s_t) = \alpha\mathcal{R}_{\text{eff}}(t)$ as a state dependent baseline. It is a foundational result of the policy gradient theorem that subtracting a state-dependent baseline $b(s_t)$ from the returns does not change the expectation of the gradient. Therefore,

$$\mathbb{E}[\hat{\nabla}_m J_{T,\text{total}}(\theta)] \approx \mathbb{E}[\hat{\nabla}_m J_{T,\text{ext}}(\theta)] = \nabla J_{T,\text{ext}}(\theta)$$

**Part 2: Variance Reduction.** The constants $B$ and $C$ important components in the variance of the reward-to-go estimator $Var(Q(s_t, a_t))$ in the ABC condition 8 and adapting to our formulation of the second moment of the gradient $\mathbb{E}[\|\hat{\nabla}_m J_{\text{total}}(\theta)\|^2]$ will affect values of $B$ and $C$ in the bound of the assumption. Thus we focus on providing an argument to reduce this. Essentially,

$$Var(Q_{\text{total}}) = Var(Q_{\text{ext}} - b(s_t))$$

From the literature, we know that the optimal baseline that minimizes variance is the **value function** itself. **Our term $-\mathcal{R}_{\text{eff}}(t)$ is designed as a proxy for the value function**: it is low when far

from the goal (low expected return) and high when near the goal (high expected return). This is essentially what we justified empirically in the correlations plot 6 of section 5.2 where we showed a strong correlation between $-\mathcal{R}_{\text{eff}}(t)$ with the value function.

Using the standard definition of $Var(X - Y) = Var(X) + Var(Y) - 2Cov(X,Y)$, we can see that:

$$Var(Q_{\text{ext}} - b(s_t)) = Var(Q_{\text{ext}}) + Var(b(s_t)) - 2 \times \rho \times \sqrt{Var(Q_{\text{ext}})} \times \sqrt{Var(b(s_t))} \quad (6)$$

where $\rho = \text{Correlation}(Q_{\text{ext}}, b(s_t))$
As discussed in our implementation (sections A.5 and A.6), we have $r_{int}(t) = \mathcal{O}(r_{ext}(t))$ and thus within reasonable approximations we consider $Var(b(s_t)) = \mathcal{O}(Var(Q_{\text{ext}}))$. This leads to the

$$Var(Q_{\text{ext}} - b(s_t)) = \mathcal{O}(Var(Q_{\text{ext}})(2 - 2\rho))$$

The larger this correlation coefficient $\rho$, the more improvement we expect in the sample complexity and thereby the convergence.
This formulation thus leads to a direct reduction in the variance-dependent constants $B$ and $C$ in assumption 8 as $B_{\text{total}} = \mathcal{O}(B_{\text{ext}}(2 - 2\rho))$ as well as $C_{\text{total}} = \mathcal{O}(C_{\text{ext}}(2 - 2\rho))$ , which even with a relatively weaker correlation of $\rho \geq 0.5$ leads to improvements. **Especially in the sparse reward settings where $Q_{ext}$ has high variance, a high correlation $\rho$ leads to a substantial variance reduction.** $\qquad \square$

Plugging the above results in Thereom 2 and noting that in sparse reward settings the terms $B$ as well as $\frac{2C}{\epsilon^2}$ in the complexity bound are the bottleneck, we have the following version of the improved sample complexity bound:

$$U_{\text{total}} \geq \frac{12\delta_0 G}{\epsilon^2} \cdot \max \left\{ B_{\text{total}}, \frac{12\delta_0 A_{\text{total}}}{\epsilon^2}, \frac{2C_{\text{total}}}{\epsilon^2} \right\} \quad (7)$$

where $U_{\text{total}}$ are the number of gradient iterations in our case.
Even with $A_{\text{total}} \approx A_{\text{ext}}$ ; to obtain an $\epsilon$-accurate solution, ie,

$$U_{\text{total}} \approx U(2 - 2\rho)$$

This justifies the faster convergence we observe empirically in section 5.2 of the main paper.

**Remark 7.** *We note that in the implementation of PPO algorithm, for which we use the above policy gradient bounds as a proxy, we only learn to approximate the value function $V^\pi$ via a neural network and not the $Q$ function as discussed in the proof A.1.4 above. Since we are dealing with environments where the agent and object dynamics are continuous, we expect the value function and $Q$ function exhibit high correlation (following works such as Nachum et al. (2018) that focus on gaussian smoothed version of the $Q$ function), ie, the states with very high value function with high probability have actions leading to high discounted cumulative rewards. Thus the plots provided in section 5.2 serve again as a proxy for what the correlation of the baseline $-\alpha\mathcal{R}_{eff}$ looks w.r.t the $Q$ function.*

## A.2 EXPERIMENTAL DETAILS

### A.2.1 AGENT

We use the Point agent from the Safety Gym similar to (Zubia et al., 2025) which has 12 dimensions (using the notation $L_{\mathcal{A}}(t)$ at the $t^{th}$ timestep for its vector):

- Accelerometer in $(-\infty, \infty)^3$, for measuring the acceleration in $m/s^2$

- Velocimeter in $(-\infty, \infty)^3$, for measuring the velocity of the agent in $m/s$

- Gyroscope in $(-\infty, \infty)^3$, for measuring the angular velocity in $rad/s$

- Magnetometer in $(-\infty, \infty)^3$, for measuring the magnetic flux in Wb

A.2.2 THE OBSERVATION VECTOR

The observation or state vector $s_t$, which we will interchangeably refer to as $L_{obs}(t)$ to characterize the LiDAR, includes the goal as well as all the other objects and is environment specific. The dimensionality of this vector is defined by the number of LiDARs used for each type of object as well as the goal. We have used 32 LiDARs for each object category in this work. As provided in the implementation of the Safety-gym library, the values of the LiDAR are normalized between 0 and 1 and they represent the *vicinity* of the objects w.r.t the agent. If any of the LiDARs record some object $o_i$ being extremely close to $\mathcal{A}$, the corresponding value $\rightarrow 1$, whereas for faraway readings the values $\rightarrow 0$.

**Practicality of Using LiDAR data ($L_{obs}$):** Before providing further details, we highlight the core reasons for using the LiDAR information as the environment state $L_{obs}(t)$ or $s_t$. In most real-world scenarios the agents are mounted with LiDARs to precisely assess the vicinity, (Choi et al., 2024; Zou et al., 2024), and training RL agents using LiDAR information has become mainstream in many recent works (Zubia et al., 2025; Miera et al., 2023; Xu et al., 2025). This has happened since the LiDARs measurements are often better than other mechanisms such as sonar, radar, cameras etc (Li & Xu, 2024; Li & Ibanez-Guzman, 2020). Other example include the entire self-driving industry (Waymo, Cruise), humanoid startups such as Boston Dynamics etc which all mount sensors, most often LiDAR, on the agents.

To generalize further to broader sensory data input, even the popular contemporary RL benchmarks such as CALVIN (Mees et al., 2022), LIBERO (Liu et al., 2023a) etc, utilize sensory information.

**Environment Specifications:**

1. Navigation: For the navigation environment, along with the goal object there are obstacles which consist of - gremlins, hazards and buttons in the environment. There is a specific negative cost associated with each of obstacle categories across difficulty levels. These negative costs are again very sparse as collisions occur rarely.

   (a) Difficulty Level 0 : consists of LiDARs for the four buttons which is a 32 dimensional vector and another set of LiDARs corresponding to the 32 dimensions for the goal object (note the goal is mounted on one of the buttons but contains a separate set of LiDARs for its tracking). Thus, $L_{obs}(t) \in [0,1]^{64}$. There are no gremlins and hazards in this environment.

   (b) Difficulty Level 1: consists of LiDARs for the four buttons which is a 32 dimensional vector, LiDARs for the 32 dimensions for the goal object, LiDAR for the four gremlins which is also a 32 dimensional vector and lastly the LiDARs for the four hazards which is again a 32 dimensional vector. Thus, $L_{obs}(t) \in [0,1]^{128}$.

   (c) Difficulty Level 2: consists of the same setting as Level 1, but this contains 6 gremlins and 8 hazards instead of 4 each in Level 1, thus making it significantly more challenging. Note that $L_{obs}(t) \in [0,1]^{128}$ since each of the 32 dimensional LiDAR vector corresponding to either type of the objects remain fixed, but the specific type of object LiDARs are now tracking many more those objects, as in the case of gremlins and hazards increasing in number at this difficulty level 2.

2. Building: In this environment, along with the goal object, there are multiple machines, other robots and risk areas which are the obstacles the agent needs to avoid. Similar to above, there is a negative cost associated with the obstacle categories across difficulty levels

   (a) Difficulty Level 0 : consists of LiDARs for the four machines which is a 32 dimensional vector and another set of LiDARs corresponding to the 32 dimensions for the goal object, this $L_{obs}(t) \in [0,1]^{64}$. There are no robots and risk areas in this environment.

   (b) Difficulty Level 1: consists of LiDARs for the four machines which is a 32 dimensional vector, LiDARs for the 32 dimensions for the goal object, set of LiDARs for the four robots which is also a 32 dimensional vector and lastly the LiDARs for the four risk areas which is again a 32 dimensional vector. Thus, $L_{obs}(t) \in [0,1]^{128}$.

   (c) Difficulty Level 2: consists of the same setting as Level 1, but this contains 6 robots and 8 risk areas instead of 4 each in Level 1, thus significantly increasing the difficulty of navigation, $L_{obs}(t) \in [0,1]^{128}$.

3. Fading: The Fading environment poses a unique challenge in that the goal disappears linearly after 150 steps of the environment's refresh. The environment contains the goal object which the agent needs to navigate to, along with obstacles termed as hazards and vases. The negative cost is again provided per category.

    (a) Difficulty Level 0: consists of LiDARs for the goal object with $L_{obs}(t) \in [0,1]^{32}$ without any obstacles.

    (b) Difficulty Level 1: consists of the usual 32 LiDARs for the goal object, 32 LiDARs corresponding to the eight hazards and 32 LiDARs corresponding to the vase. Thus $L_{obs}(t) \in [0,1]^{96}$

    (c) Difficulty Level 2: consists of the usual 32 LiDARs for the goal object, 32 LiDARs corresponding to the ten hazards and 32 LiDARs corresponding to ten vases. Thus $L_{obs}(t) \in [0,1]^{96}$, with significantly more obstacles to avoid disappearing goal object.

## A.3 VISUALIZATIONS

**Environments:** A visualization representation of all these combinations of environments with varying difficulty levels is provided in figures 7, 8 and 9 respectively.

**Goal Not Captured at certain timestep:** Our definition of $r_{int}(t)$ from section 3.1 characterizes the visibility of goal and a significant penalty of $-\beta$ when its not captured on Agent's LiDAR. We provide a sample visualization of a case where the agent drifts far off in the environment in figure 10 such that the goal is outside the sensing range and thus the indices corresponding to the goal in the observation vector become zero. These high negative intrinsic reward situations provide a strong signal to the agent to learn a better policy and we empirically show that the learned policies have much better navigation during evaluation in section A.8.2.

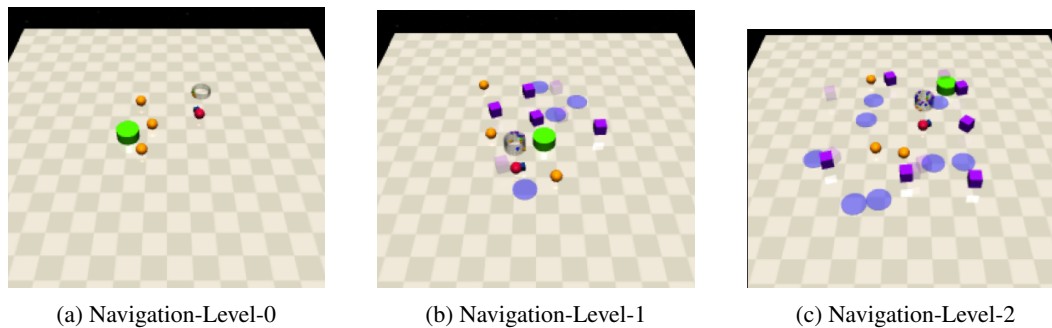

        (a) Navigation-Level-0         (b) Navigation-Level-1         (c) Navigation-Level-2

Figure 7: Visualization of the Navigation Environment with Level-0 (left figure), Level-1 (center figure) and Level-2 (right figure), where the *Red spherical ball with a pointed blue cube* is the agent with the LiDARs mounted over it and the goal is the green colored cylinder.

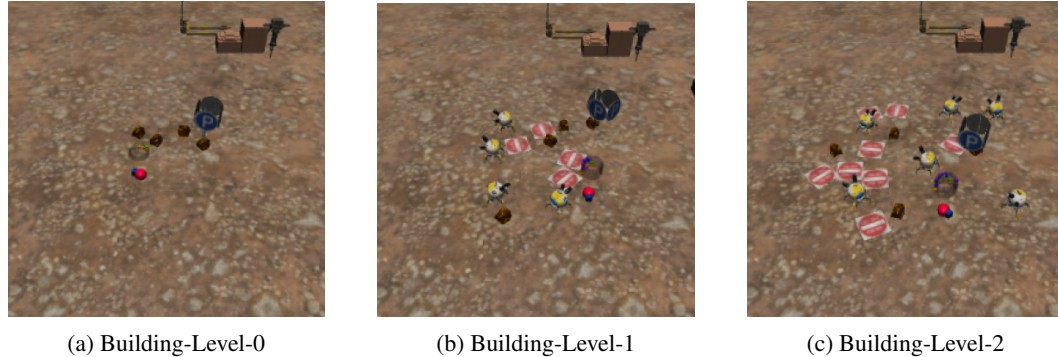

(a) Building-Level-0      (b) Building-Level-1      (c) Building-Level-2

Figure 8: Visualization of the Building Environment with Level-0 (left figure), Level-1 (center figure) and Level-2 (right figure), where the agent needs to operate various machines with the Parking symbol being the goal machine.

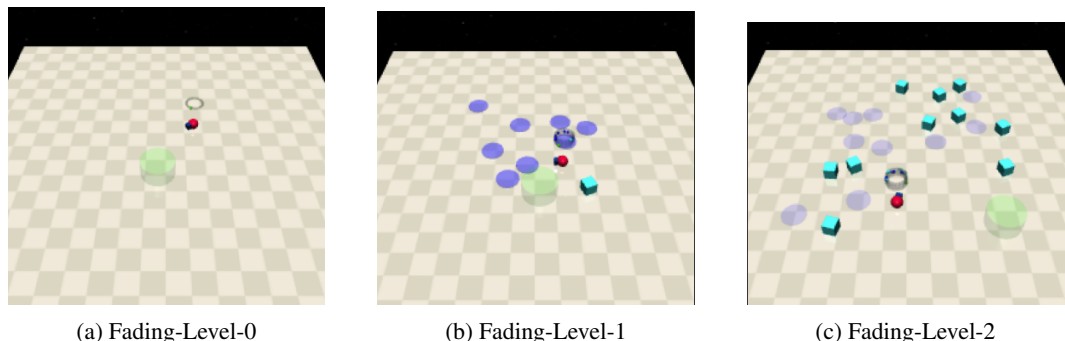

(a) Fading-Level-0      (b) Fading-Level-1      (c) Fading-Level-2

Figure 9: Visualization of the Fading Environment with Level-0 (left figure), Level-1 (center figure) and Level-2 (right figure). Notice how the goal appears faded and eventually vanishes.

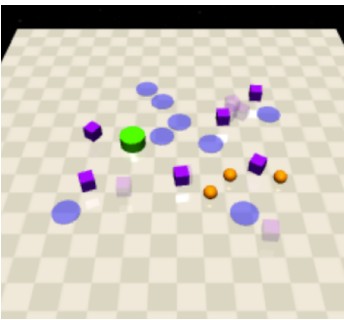

Figure 10: An example from the Navigation Environment where the agent wanders off far away and thus the goal is not captured on the agent's LiDAR. The agent is thus also not captured in the camera angle.

A.4  GRAPH CONSTRUCTION ALGORITHM

---

**Algorithm 1** Graph Construction from LiDAR Data

---

**Input:** LiDAR observation vector $L_{obs} \in \mathbb{R}^N$ (environment specific), LiDAR threshold $\tau$
**Output:** Graph $\mathcal{G}_t = (V_t, E_t, W_t)$ with weighted-undirected edges

1: **Initialize:**
2: Graph $\mathcal{G}_t$ with empty node set $V_t$, edge set $E_t$, and weight set $W_t$.
3: Create central agent node $v_{\mathcal{A}}$ and add to $V_t$.
4: Fetch the partitions of $L_{obs}$ for each category (object type): $L_{\mathcal{S}^1}, \cdots, L_{\mathcal{S}^i}, \cdots, L_{\mathcal{S}^P}, L_{goal}$, where $i \in \{1 \le i \le P\}$, $P$ is number of categories (excluding Agent and Goal).

5:                                       ▷ 1. Generate nodes for each category via LiDAR segments using algo 2
6: $V_{clusters} \leftarrow \{\text{SegmentToNodes}(L_{\mathcal{S}^1}), \cdots, \text{SegmentToNodes}(L_{\mathcal{S}^P}), \text{SegmentToNodes}(L_{goal})\}$

7: Add all nodes from $V_{clusters}$ to $V_t$.

8:  ▷ 2. Connect agent to each node and also Create fully connected subgraphs within each cluster
9: **for** each cluster $V_C$ in $\{V_{\mathcal{S}^1}, \cdots, V_{\mathcal{S}^P}\}$ **do**
10:     **for** each node $v_i$ in $V_C$ **do**
11:         Add edge $(v_{\mathcal{A}}, v_i)$ to $E_t$ with weight $w_{\mathcal{A},i}$ equal to agent-object vicinity
12:     **end for**
13:     **for** each pair of distinct nodes $(v_i, v_j)$ in $V_C$ **do**
14:         Add edge $(v_i, v_j)$ to $E_t$ with weight $w_{ij}$ equal to inter-object vicinity
15:     **end for**
16: **end for**

17:                                                       ▷ 3. Compute cluster representatives
18: $V_{reps} \leftarrow \emptyset$
19: **for** each cluster $V_C$ in $V_{clusters}$ **do**
20:     Calculate representative node $v_{rep}$ (e.g., degree central node) for $V_C$ and add to $V_{reps}$.
21: **end for**

22:                                              ▷ 4. Connect all cluster representatives to each other
23: **for** each pair of distinct representatives $(v_{rep\_i}, v_{rep\_j})$ in $V_{reps}$ **do**
24:     Add edge $(v_{rep\_i}, v_{rep\_j})$ to $E_t$ with weight based on vicinity.
25: **end for**

26: **return** $\mathcal{G}_t$

---

---

**Algorithm 2** Segment to Nodes Function

---

1: **function** SEGMENTTONODES($L_{segment}$)
2:     $V_{nodes} \leftarrow \emptyset$                   ▷ Initialize an empty set for nodes in this segment
3:     $current\_node\_readings \leftarrow []$
4:     **for** $i = 1$ to length($L_{segment}$) **do**
5:         $d_i \leftarrow L_{segment}[i]$               ▷ Current LiDAR reading (vicinity)
6:         **if** $d_i > 0$ **then**
7:             **if** is empty($current\_node\_readings$) or $|d_i - $ last($current\_node\_readings$)$| < \tau$
    **then**
8:                 Append $d_i$ to $current\_node\_readings$
9:             **else**
10:                Create node $v$ from $current\_node\_readings$; Add $v$ to $V_{nodes}$
11:                $current\_node\_readings \leftarrow [d_i]$        ▷ Start a new node
12:             **end if**
13:         **else if** is not empty($current\_node\_readings$) **then**
14:             Create node $v$ from $current\_node\_readings$; Add $v$ to $V_{nodes}$
15:             $current\_node\_readings \leftarrow []$       ▷ Reset on zero reading
16:         **end if**
17:     **end for**
18:     **if** is not empty($current\_node\_readings$) **then**        ▷ Add the last processed node
19:         Create node $v$ from $current\_node\_readings$; Add $v$ to $V_{nodes}$
20:     **end if**
21:     **return** $V_{nodes}$
22: **end function**

---

### A.4.1   SAMPLE GRAPH VISUALIZATION CONSTRUCTED VIA ALGORITHM 1

A visual representation of the graph that is constructed via algorithm 1 is provided in figure 11 along with the respective environment state in figure 12

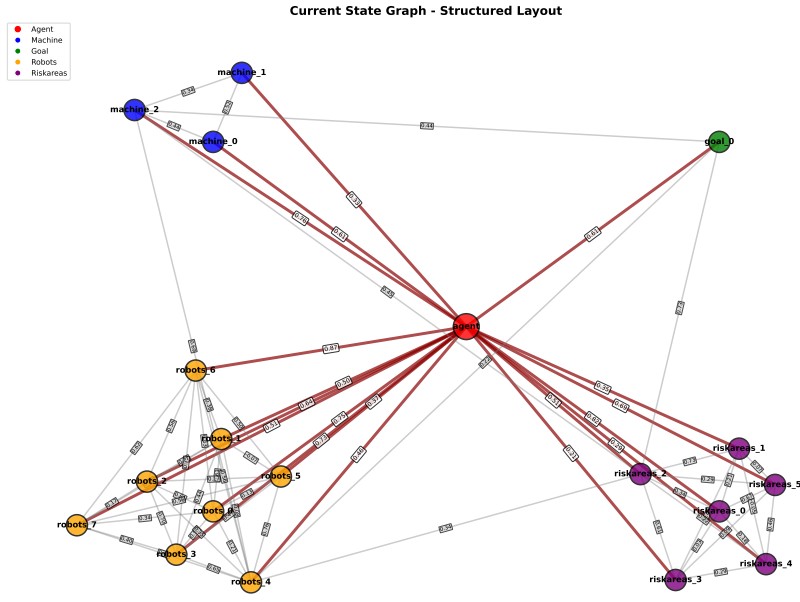

Figure 11: A sample visualization of the generated graph on the Building-Level-2 environment. The edge weights here describe the vicinity of the objects as discussed in the graph construction algorithm. The dark red edges are emphasized to show the edges from agent node to all other nodes, and remaining edges shown in lighter color.

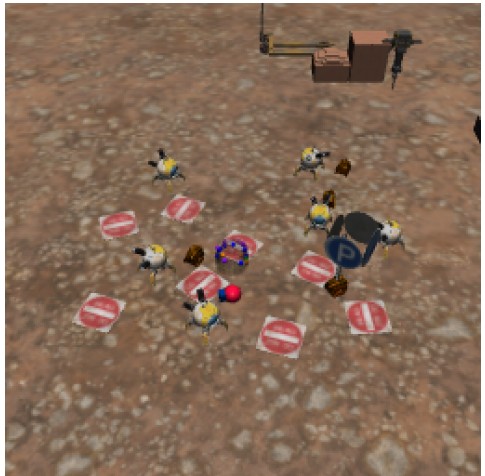

Figure 12: The environment state corresponding to the graph in figure 11

## A.5 IMPLEMENTATION AND HYPERPARAMETER DETAILS

### A.5.1 GRAPH CONSTRUCTION HYPERPARAMETERS

Since the construction of $\mathcal{G}_t$ in algorithm 1 is the core of our method, we discuss the corresponding design and hyperparameter choices as follows and explicitly detail these in table 1:

1. The value of LiDAR reading difference threshold $\tau$, which is used to generate the nodes, is a fairly non-trivial problem which depends on many factors of the environment dynamics. We first obtained an approximate range for $\tau$ analytically, described in section A.5.3. Using this range, we performed a sensitivity analysis over a set of values (results in section A.9). Aggregating everything, we select $\tau = 0.1$. Additionally, we emphasize that the consistent choice of $0.1$ also eases the setting of hyperparameters and reproducibility across environments.

2. Graph Laplacian is **un-normalized**. Consider a simple example where there we have two graphs $\mathcal{G}_1$ and $\mathcal{G}_2$ where $\mathcal{G}_2$ has the exact same connectivity as $\mathcal{G}_1$ but the edge weights of $\mathcal{G}_2$ are scaled by some arbitrary factor $b$, ie the adjacency matrix of $A_{\mathcal{G}_2} = b \cdot A_{\mathcal{G}_1}$ (where $b$ is multiplied to all elements of the adjacency matrix). Now the *Normalized* graph laplacians of both the graphs are exactly the same, $L_{\mathcal{G}_2} = L_{\mathcal{G}_1}$, due to the normalization property by the diagonal degree matrix. This has a counter effect as it completely ignores the environment state and location of the goal w.r.t the agent which we also want to encode in $r_{int}(t)$ in the first place. On the other hand, the *Un-normalized* laplacians provide us the form $L_{\mathcal{G}_2} = b \cdot L_{\mathcal{G}_1}$, retaining the changes of all other objects w.r.t the agent.

3. For the choice of graph connectivity (structure) described in algorithm 1, note that simply the possible edge connections (ignoring the edge weights) are combinatorial and thus an informed design decision has to be made to construct the graph $\mathcal{G}_t$ from $L_{obs}(t)$. We reiterate the algorithm here along with the rationale behind these decisions:

   (a) Using each category cluster $V_C$, a star graph is created where the agent node is connected to every other node in the graph. This is done since in any environment the agent is the core node w.r.t which we obtain information about the surroundings.

   (b) Then we construct complete graphs within each $V_C$ in order to symbolize a strong notion of object category (described in detail in section A.2 for all environments). Another straightforward way to do is to introduce a different edge type for each object, however such a construction makes the graph more complicated and usually requires further decomposition in order to compute any property of the graph. Thus we restrict ourselves to a unified edge type identified by its node-edge tuple $(i, j, w_{ij})$. Two strong reasons led to this choice - (i) the number of nodes for each object type are limited and thus even for a complete graph the subgraph size remains manageable, (ii)

(Li & Zehmakan, 2023) describes the ease of obtaining node centrality (see next point for this) in a complete graph.

(c) We then obtain the degree based central nodes, also called representatives, in $V_{reps}$ and connect those amongst each other making another subgraph. This type of subgraph is generally small in most scenarios of interest as we are considering different types of objects (in an environment which contains too many types, one can perform some form of clustering/grouping) and is thus manageable. While the construction of this subgraph can be ignored, this however leads to a relatively trivial graph which only factors in the agent to object nodes, and nodes of same object type, but completely ignore the geometric locations of the object type w.r.t one another. The choice of connecting *only the central nodes of the object types* is made to restrict the graph size from becoming fully connected which increases the complexity of Laplacian Pseudoinverse computation. We have empirically justified this central node connectivity in section A.12 ablation.

(d) **Note:** We only generate 1 node for the goal object in algorithm 1. The reason is that - (i) this simplifies the calculation of $r_{int}(t)$ (in equation 2) in contrast to the case where we have multiple goal nodes, (ii) in any practical environment, by default the goal LiDAR values in $L_{obs}(t)$ are contiguous as the goal is a single object and their difference is $\ll \tau$ which creates a single node automatically.

Table 1: Our method Graph Construction Hyperparameters

| Parameter | Value |
|---|---|
| LiDAR reading difference threshold $\tau$ | 0.1 |
| Graph $\mathcal{G}_t$ Laplacian type | Un-normalized |

### A.5.2 PPO HYPERPARAMETERS

Since our method is directly based on the PPO implementation, we retain same hyperparameters for PPO, PPO with Entropy and Our Method in order to perform the direct analysis of adding our proposed $r_{int}(t)$. Hyperparameters such as learning rate, reward discounting factor $\gamma$, factor for Generalized Advantage Estimates, entropy lambda factor for PPO with entropy etc are borrowed from the recent literature (Xudong et al., 2024). Based on many works in the literature, we also utilize parallel PPO implementation - these agents all operate in independent instances of the corresponding environment in order to collect the demonstrations faster and we note that *this is NOT multi-agent RL* setting. We use a shared 2-layer MLP to feed the concatenation of the agent vector (section A.2.1) $L_{\mathcal{A}}(t)$ and the $L_{obs}(t)$ (aka the state vector $s(t)$ from section A.2.2) as input which is a common approach in the literature. Over this shared MLP backbone we have two separate heads - the first one is linear projection of size $256 \times 2$ which behaves as the **Policy Network** $\pi_\theta$ to predict the action whereas the second one is $256 \times 1$ for the **critic** for Value function approximation. As discussed in sections A.2.1 and 3.1, the action space is $[-1, 1]^2$ corresponding to the *throttle* and *steering angle* of the agent respectively which, following (Barhate, 2021), is formulated as a gaussian $\mathcal{N}(\mu_\theta(L_{obs}(t)), \sigma^2)$. The Policy Network output is this 2-dim prediction for the mean and there is a separate Pytorch trainable parameter of size 2 for the standard deviation values (therefore the standard deviation values are learned directly via gradient updates). We also discuss the evolution of this standard deviation values over the course of training in section A.13. During **training** an action is sampled at random from this gaussian whereas during **evaluation** we simply consider the mean (which is the output of the Policy Network as is) based on the original pytorch implementation. The hyperparameter values are provided in table 2

For the remaining baselines - we take source code of the Surprise RL Standard and Surprise RL Diff and train it with PPO as it is the SOTA RL algorithm. For MEGA and GCPO, we borrowed the source codes from their implementations and MEGA Github (MRL Library of the author) and GCPO Github respectively.

For the baselines using entropy loss, the entropy is calculated over the gaussian of the aforementioned action distribution following the actual implementation. The experiments were conducted on NVIDIA A100 GPUs.

Table 2: PPO algorithm hyperparameters. Note that using 20 agents in parallel simply speeds up the process for on-policy setup and all these agents operate in independently created instances of the environment.

| Parameter | Value |
|---|---|
| Learning Rate | $1e^{-4}$ |
| Batch Size | 2048 |
| Num Agents Used in Parallel | 20 |
| Num Episodes per agent | 2500 |
| Max Timesteps per episode | 1000 |
| Num Episodes between Policy Update | 2 |
| Num Epochs for update | 4 |
| Entropy Coefficient (for baselines using entropy loss) | $1e^{-2}$ |
| Shared MLP size | $[|L_{\mathcal{A}}| + |L_{obs}|, 256] \rightarrow [256, 256]$ |
| Actor size | Shared MLP $\rightarrow [256, 2]$ |
| Critic size | Shared MLP $\rightarrow [256, 1]$ |
| Discounting Factor $\gamma$ | 0.99 |
| Generalized Advantage Estimate factor | 0.95 |
| PPO clipping $\epsilon$ | 0.2 |

### A.5.3 Obtaining the range for threshold $\tau$

We first reiterate (as already mentioned in section A.5.1) that obtaining a tight range for $\tau$ is a fairly non-trivial problem that requires a deeper investigation of its own. However, here we lay out the approximations that we used to first obtain a range for $\tau$ which we then plugged in with the corresponding observations from the environment and then lastly did the sensitivity analysis to arrive at the specific value of $\tau = 0.1$ that worked well across most settings and furthermore as pointed out earlier in section A.5.1, a single value helps in reproducibility as well.

As mentioned earlier in section A.2.2, the LiDAR values denote the *vicinity* to the agent rather than the normalized distance. However for the ease of reading, we use the notion of distance below but it is trivial to switch to vicinity/closeness. We have the following steps in our derivation:

**1. Approximation of $\tau$ and its intuition:** Defining

- $R$ be the sensing (LiDAR) radius, discussed earlier in section A.1.1
- Suppose we are merging the LiDAR values (for a detailed discussion on the LiDAR values we refer the reader to section A.2.2) of two arbitrary objects $o_1$, $o_2$ using $\tau$ in the *Segment-ToNodes* function defined in algorithm 2. Then we define $\hat{d}$ as the distance from agent to the nearer of the two objects being considered (i.e. $\hat{d} = \min\{d_{\mathcal{A},o_1}, d_{\mathcal{A},o_2}\}$), with $d_{\mathcal{A},o_i}$ being agent to object $o_i$ distance
- let $r_c$ be the collision buffer term which is characterized by the agent's size, the object size(s) along with some small safety margin
- let $\psi$ be a slack parameter that we use to control the the dependence on $\hat{d}$. In other words, it defines how aggressively we combine objects in a single node as the value of $\hat{d}$ decreases

We define the upper bound on $\tau$ as a function of $\hat{d}$ as follows:

$$\tau(\hat{d}) \;<\; \frac{\min(R, d_{sep})}{R} \;=\; \frac{\min(R, r_c + slack(\hat{d}))}{R} \;=\; \frac{\min\left(R,\; r_c + \psi\left(1 - \frac{\hat{d}}{R}\right)\right)}{R}. \quad (8)$$

where $d_{sep} = r_c + slack(\hat{d})$ is the separation term depending upon $r_c$ (which becomes fixed once we know the agent and obstacle dimensions) and some slack term which is a function of $\hat{d}$. As a key property, we need this slack term to behave in a manner such that when the objects are far from the agent, we can create separate nodes in our graph and it does not have a notable impact on the $\mathcal{R}_{\text{eff}}$ value. At $\hat{d} \to 0$, the objects are fairly close to the agent $\mathcal{A}$ and thus we create a unified node in the graph, keeping a larger slack term. The choice of a linear decay formulation as $slack(\hat{d}) = \psi\left(1 - \frac{\hat{d}}{R}\right)$ is the simplest formulation satisfying the desired property, defined next.

**2. Approximation of $\psi$ in terms of agent and object velocities:** While the choice of $\psi$ again is complex, we leverage an aggregate of the velocities (exact computation discussed below) of the agent and objects

$$\alpha = \frac{V_{agg}}{V_{agg} + v_{agg}} \, R$$

where

- $v_{agg}$ = aggregate of the speed (interchangeably used for velocity when direction of motion is fixed) of the agent
- $V_{agg}$ = aggregate of the speed of the objects

Thus, when $v_{agg} >> V_{agg}$, the likelihood of the agent navigating through the space between the objects is extremely high admitting $slack(\hat{d}) \downarrow 0$ and vice-versa.

**3. Aggregating over $\hat{d}$:** While equation 8 characterizes the threshold, we still have dependency over the term $\hat{d}$. We integrate over this term by assuming that the objects $o_1$ and $o_2$ are independently located within the radius $R$ centered around the agent.

Thus the cumulative density function, $F_{\hat{d}}$, of $\hat{d} = \min\{d_{\mathcal{A},o_1}, d_{\mathcal{A},o_2}\}$ can be defined as follows:

$$F_{\hat{d}}(x) = \mathbb{P}(\hat{d} \leq x) = 1 - \mathbb{P}(d_{\mathcal{A},o_1} > x, d_{\mathcal{A},o_2} > x) \tag{9}$$

$$= 1 - (1 - F_d(x))^2 \tag{10}$$

where $F_d(x)$ is cumulative density function of an arbitrary object being located within a distance $x$ from the agent on the disc of radius $R$, ie, $F_d(x) = \frac{x^2}{R^2}$ (simply the ratio of areas). Plugging this above with the independence of the location of the objects, we obtain

$$F_{\hat{d}}(x) = 1 - \left(1 - \frac{x^2}{R^2}\right)^2$$

Since this function is differentiable, we can compute the PDF as

$$f_{\hat{d}}(x) = 4\frac{x}{R^2}\left(1 - \frac{x^2}{R^2}\right), 0 \leq x \leq R$$

The expectation $\mathbb{E}(\hat{d})$ is then

$$\mathbb{E}(\hat{d}) = \int_0^R x f_{\hat{d}}(x)dx = \frac{8}{15}R \tag{11}$$

**4. Putting it all together: aggregate value of $\tau$:** Using the derivations above and putting them in equation 8, we obtain an aggregate of $\tau$ as

$$\tau < \min\left(1, \frac{r_c + \frac{V_{agg}}{V_{agg}+v_{agg}} R \left(1 - \frac{8R/15}{R}\right)}{R}\right) \tag{12}$$

$$= \min\left(1, \frac{r_c + \frac{V_{agg}}{V_{agg}+v_{agg}} \frac{7}{15}R}{R}\right). \tag{13}$$

Numerical approximation:

1. typically $r_c << R$ and thus we can ignore this term

2. to obtain a reasonable estimate of $V_{agg}$ and $v_{agg}$, we compute the median values via a monte carlo estimate as follows:

   (a) We considered $10,000$ arbitrary timesteps independently for each of the 3 environments and the respective 3 difficulty levels

   (b) for each timestep $t$ of these $10000$ rollouts, we obtained the median of the absolute value differences between the LiDAR vector values at $t$ and corresponding $t+1$. This is a loose approximation since these LiDAR values are w.r.t the agent, but one that suffices as the agent and the objects have restricted motion at any particular timestep. For the agent, we already have the estimate of the velocities using its velocimeter as discussed in section A.2.1

   (c) obtain the median values of the distances for the objects and the median values for the velocities of the agent respectively

   (d) compute the ratio $\frac{V_{agg}}{V_{agg}+v_{agg}}$ (as in our case velocity is being calculated as the distance in a particular direction over 1 unit of time)

   (e) The rationale for using median is the same as earlier experiments that the mean values are extremely noisy (with the standard deviation being higher than mean in a skewed distribution where the absolute values are bounded below by $0$). Thus, we rely on the median in this estimate.

Plugging all the values, we obtained $\tau < \min(1, 0.124) = 0.124$. We then performed the sensitivity analysis below this threshold value, with more details in section A.9.

### A.6 SETTING VALUES OF $\beta$ AND $\alpha$

Corollary A.1.3 provides practical implementation choices for the values of $\beta$ and $\alpha$ in terms of the maximum absolute extrinsic reward and the maximum graph connectivity change respectively. We discuss the selection below and note the values explicitly in table 3

**Selecting $\beta$:** Based on the code of the Safety Gym library, we noted that the maximum absolute value of the extrinsic reward is $< 1$ and thus factoring in the constant $C$, we performed some initial experimental checks and select $5$ as a penalty factor to keep, which is sufficiently but not extremely large, to steer the agent for goal visibility. Values greater than $1$ worked reasonably well and $5$ is primarily a design choice.

**Selecting $\alpha$:** We consider each of the components $\max_{s,a}|r_{\text{ext}}(s,a)|$, $\delta_{\max}$ and $C_1$ separately:

1. from the empirical details of section 5, we have $\max_{s,a}|r_{\text{ext}}(s,a)| < 10$ as we scaled it with a factor of $10$

2. Note that $\delta_{max}$ derived in lemma A.1.3 $= \mathcal{O}(k)$ , where $k$ defines maximum number of node changes in the graphs per timestep. Based on the environment specifications from section A.2. As discussed previously, the event $\mathcal{E} : \{k(\Delta t) \gg n_{\min}\}$ almost never occurs, and we thus retain $k = 1$

3. $C_1$ as the lower bound of $C(t) = \frac{1}{\kappa(t)^2}\langle e_{\mathcal{A}} - e_g, v_2(t)\rangle^2$ , which we empirically observed to be $< 0.837$

Putting it all together, we approximate $\alpha = 10$. If we plug in the numbers, we obtain $\frac{10}{0.837} = 11.94$. We however, to again focus on simplicity and consistency of the same hyperparamter across the environments, chose the value of $10$, similar to the selection of $\tau$ earlier.

Table 3: The values of $\alpha$ and $\beta$ in the implementation of Our method.

| Parameter | Value |
|---|---|
| $\beta$ penalty factor to enforce goal visibility | 5 |
| $\alpha$ factor for scaling $r_{int}$ | 10 |

**A Note on Robustness:** It is critical to note that these single, principled values ($\beta = 5, \alpha = 10$) were used consistently across **all 9 environments** (all 3 tasks and all 3 difficulty levels) without

any per-environment tuning. This stands in stark contrast to methods that require intensive, per-task hyperparameter sweeps and strongly demonstrates the robustness and generality of our theoretically-grounded selection process.

## A.7 EVOLUTION OF $r_{int}$ DURING NAVIGATION

We plot the cumulative intrinsic reward $\sum_{t=1}^{i} r_{int}(t), i \in [T^e]$ where $T^e$ is the length of the corresponding episode $e$ being evaluated upon, at the median along with the $25^{th}$ and $75^{th}$ percentiles with shaded region over these 1000 (200 per trained model and 5 models with random seeds) evaluation episodes. The aim of this experiment is to assess how the $r_{int}$ evolves as the agent navigates the environments post the policy training. Since $r_{int}$ fluctuates at every time step depending upon the current state, we visualize the cumulative value. The results are plotted in figures 13, 14 and 15 for the environments respectively.

Note that there are many timesteps during a trajectory rollout where $r_{int}(t)$ is negative for some timestep $t$ but the agent is still closer to the goal. This happens due to a more intricate dynamics where all the other objects are also navigating in the environment and this impacts the edge weights in $\mathcal{G}_t$. The effects of this can be noticed in the plots for difficulty level 2 primarily. We discuss this in more detail with a thorough quantitative analysis in next section A.7.1.

Another noteworthy aspect is the difficulty of navigating in the environment can also be assessed via these plots. As observed in the main section result figures 2, 4 and 3, difficulty level 2 (and level 1 to some extent), which are harder for the agent to navigate in due to presence of large number of randomly moving obstacles, its corresponding $r_{int}$ also exhibits fluctuations.

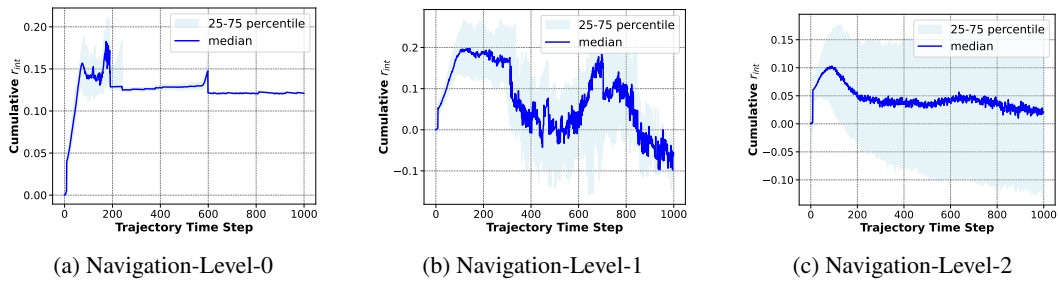

|  |  |  |
|---|---|---|
| (a) Navigation-Level-0 | (b) Navigation-Level-1 | (c) Navigation-Level-2 |

Figure 13: Visualization of the Cumulative $r_{int}$ across the timesteps for the Navigation Environment with Level-0 (left figure), Level-1 (center figure) and Level-2 (right figure)

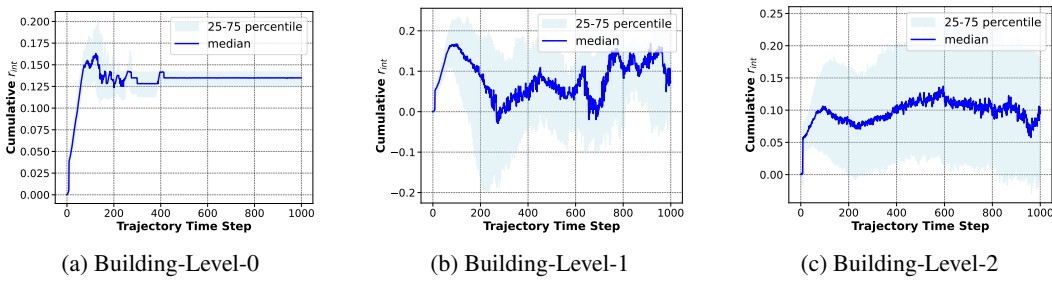

|  |  |  |
|---|---|---|
| (a) Building-Level-0 | (b) Building-Level-1 | (c) Building-Level-2 |

Figure 14: Visualization of the Cumulative $r_{int}$ across the timesteps for the Building Environment with Level-0 (left figure), Level-1 (center figure) and Level-2 (right figure)

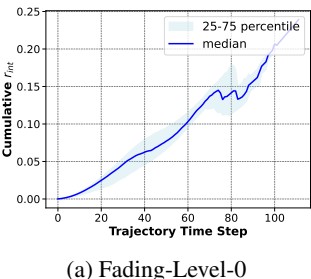 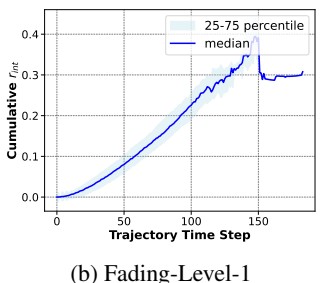 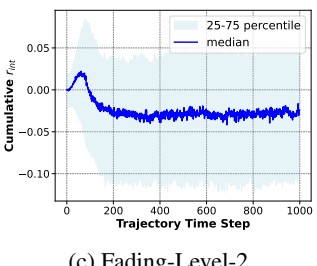

(a) Fading-Level-0        (b) Fading-Level-1        (c) Fading-Level-2

Figure 15: Visualization of the Cumulative $r_{int}$ across the timesteps for the Fading Environment with Level-0 (left figure), Level-1 (center figure) and Level-2 (right figure)

### A.7.1 CONFIGURATIONS FOR NEGATIVE $r_{int}(t)$ BUT IMPROVED NAVIGATION TO GOAL

As mentioned in the previous section, while plotting $r_{int}$ (section A.7), there can be configurations where $\mathcal{R}_{\mathrm{eff}}(t) - \mathcal{R}_{\mathrm{eff}}(t+1) \leq 0$ but the agent navigates closer to the goal. Note that such situations occur since the computation of $\mathcal{R}_{\mathrm{eff}}$ is non-trivial w.r.t to the distance of agent to the goal, as it incorporates the entire observation space captured on its LiDAR $L_{obs}(t)$ and $L_{obs}(t+1)$. To quantify such situations, we perform the following experiment:

1. Load each of the $t$ trained policies
2. Roll each policy for 20 episodes, thus 100 ($20 \times 5$) arbitrary rollouts in total incorporating for the randomness
3. Obtain the count of the timesteps at which the aforementioned situation of negative $r_{int}$ but agent navigating closer to the goal happened. Lets call this count $J_{r_{int}}^{neg}$
4. Obtain the total number of timesteps that the agent navigated across all episodes, call this count $J$
5. Compute the ratio

The outcomes mentioned in table 4 justify the fluctuations in the plots of the $r_{int}$ and how it affects the monotonicity of even the cumulative reward.

Table 4: Fraction of times when $r_{int} < 0$ but agent moves closer to the goal, in order to justify the rationale behind plotting the cumulative $r_{int}$ instead of raw values in section A.7

| Environment Name | $J_{r_{int}}^{neg}$ | $J$ | Ratio $\frac{J_{r_{int}}^{neg}}{J}$ |
|---|---|---|---|
| Navigation-level-0 | 501 | 5977 | 0.0838 |
| Navigation-level-1 | 2220 | 10293 | 0.2157 |
| Navigation-level-2 | 12138 | 51186 | 0.2371 |
| Building-level-0 | 627 | 6755 | 0.0928 |
| Building-level-2 | 3132 | 16008 | 0.1956 |
| Building-level-2 | 9472 | 37007 | 0.2560 |
| Fading-level-0 | 84 | 5123 | 0.0164 |
| Fading-level-1 | 1928 | 7047 | 0.2736 |
| Fading-level-2 | 12078 | 51846 | 0.2329 |

### A.8 BRIDGING THE RESULTS FROM LEMMA 1 AND THEOREM 1 WITH EMPIRICAL OBSERVATIONS

### A.8.1 EMPIRICAL VERIFICATION OF BOUNDED GRAPH CONNECTIVITY

We proved in lemma 1 that the change in $\mathcal{R}_{\mathrm{eff}}$ and $\kappa$ have opposite gradient signs over time. Earlier we showed that generally $\mathcal{R}_{\mathrm{eff}}$ tends to decrease over time, and as such, we expect $\kappa(\mathcal{G}_t)$ to increase. We also claimed in the first part of theorem 1 that the expected graph connectivity at timestep $t+1$ is lower bounded via $\mathbb{E}_{\tau \sim \pi^*}[\kappa(\mathcal{G}_{t+1})|\mathcal{G}_t] \geq \kappa(\mathcal{G}_t) - \delta_{\max}$ where the $\delta_{\max}$ is from lemma A.1.3.

To justify both of these empirically, we we provide the plots for the absolute difference of $\kappa(\mathcal{G}_{t+1})$ and $\kappa(\mathcal{G}_t)$ across the timesteps at the median along with the $25^{th}$ and $75^{th}$ percentile bands across the environments and difficulty levels in figures 16, 17 and 18. The following important observations can be made:

1. as argued from the lemma 1, we expect the value of $\kappa$ to increase over time. This essentially leads to larger absolute differences which can observed in various plots.

2. we notice various arbitrary fluctuations in the plots. These values typically characterize non-trivial changes to the graph structure, which are also reflected in $\delta_{\max}$ dervied in lemma 2. Using $\delta_{\max} = \mathcal{O}(\|\|)$ for $k = 1$, clearly holds.

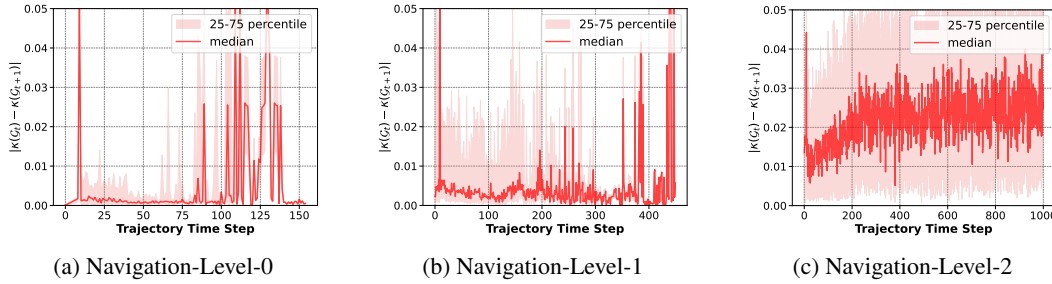

| (a) Navigation-Level-0 | (b) Navigation-Level-1 | (c) Navigation-Level-2 |

Figure 16: The absolute value of the difference of second eigenvalues, ie $|\kappa(\mathcal{G}_t) - \kappa(\mathcal{G}_{t+1})|$ across the Navigation environment.

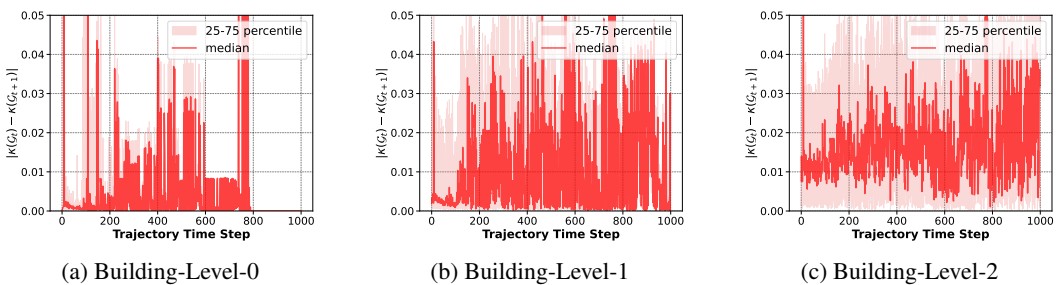

| (a) Building-Level-0 | (b) Building-Level-1 | (c) Building-Level-2 |

Figure 17: The absolute value of the difference of second eigenvalues, ie $|\kappa(\mathcal{G}_t) - \kappa(\mathcal{G}_{t+1})|$ across the Building environment.

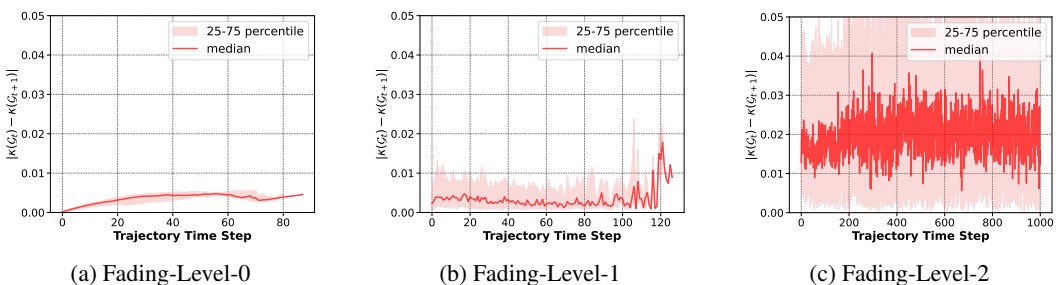

| (a) Fading-Level-0 | (b) Fading-Level-1 | (c) Fading-Level-2 |

Figure 18: The absolute value of the difference of second eigenvalues, ie $|\kappa(\mathcal{G}_t) - \kappa(\mathcal{G}_{t+1})|$ across the Fading environment.

### A.8.2 EMPIRICAL VERIFICATION OF GOAL VISIBILITY MAINTENANCE

We also argue that a well trained (otherwise optimal) policy will learn to maintain a high goal visibility - (i) in order to maximize the expected reward and (ii) to avoid the large negative penalty

of $-\beta$. We demonstrate that this indeed happens and during evaluation the 5 trained policies indeed maintain a much higher goal visibility as shown in table 5 across all environments and difficulty levels in comparison to randomly initialized policies.

Table 5: Percentage of timesteps in which the goal is **visible** on the agent's observation $L_{obs}$ during the evaluation in contrast to a randomly initialized policy.

| Environment Name | Randomly Initialized Policy | Trained Policy |
|:---:|:---:|:---:|
| Navigation-level-0 | 50.26% | 99.59% |
| Navigation-level-1 | 48.61% | 99.35% |
| Navigation-level-2 | 56.39% | 98.99% |
| Building-level-0 | 85.33% | 99.48% |
| Building-level-1 | 81.00% | 99.27% |
| Building-level-2 | 77.12% | 99.09% |
| Fading-level-0 | 94.88% | 99.92% |
| Fading-level-1 | 88.60% | 99.67% |
| Fading-level-2 | 86.90% | 99.21% |

## A.9 SENSITIVITY ANALYSIS OF GRAPH CONSTRUCTION THRESHOLD $\tau$

One of the core components of our method is the threshold $\tau$ which is used in the clustering of elements of the LiDAR array to construct nodes in the graph as discussed in algorithm 1. Since, we numerically attained $\tau < 0.124$ earlier, we vary $tau \in \{0.08, 0.09, 0.1, 0.11, 0.12\}$ and evaluate the corresponding policies. Since the values in this set are relatively close, we also expect less susceptibility. The sensitivity analysis plots are provided in figure 19 (all the computations are done in the same manner as discussed in Evaluation Methodology of section 5). While we do notice minor fluctuations in the results and there is no unified value of $\tau$ that works the best across all types of evaluation metrics and environments, $\tau = 0.1$ still provides a great parameter setting and can be used for all difficulty levels and environments, thus significantly reducing the hyperparameter search workload mentioned previously in section A.5.1.

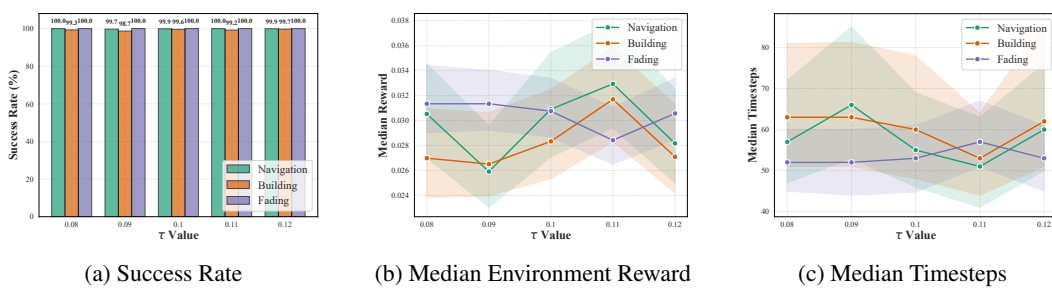

(a) Success Rate      (b) Median Environment Reward      (c) Median Timesteps

Figure 19: Success Rate, Median Timesteps and Median Normalized Reward across the environments for varying values of the graph construction hyperparameter $\tau$ in algorithm 1

## A.10 COMPLEXITY ANALYSIS AS A FUNCTION OF GRAPH SIZE

We provide a formal analysis of the computational complexity of our method, which is incurred at each timestep. The total per-step cost is the sum of (1) graph construction and (2) effective resistance computation.

1. **Graph Construction (Algorithm 1):** The complexity of constructing $\mathcal{G}_t$ is highly efficient. The `SegmentToNodes` function (Algorithm 2) iterates through the LiDAR observation vector ($L_{obs}(t)$), say of size $N$, which has a complexity of $\mathcal{O}(N)$. The subsequent connection steps (lines 9-25 in Algorithm 1) iterate through the nodes $n = |V_t|$ and object categories $P$. In the worst case, this involves creating a complete graph within clusters, resulting in a complexity of $\mathcal{O}(n^2)$. Since the dependence on $N$ is linear and $n$ is designed to be small, the total graph construction cost, which is additive, of $\mathcal{O}(N + n^2)$ is not large.

2. **Effective Resistance Computation (Eq 2):** This is the primary computational bottleneck of our intrinsic reward. The calculation, as defined in **Eq 2**, requires the Moore-Penrose pseudoinverse, $L_t^+$, of the $n \times n$ graph Laplacian $L_t$. Standard numerical methods for computing the pseudoinverse (e.g., via Singular Value Decomposition) have a computational complexity of $\mathcal{O}(n^3)$, where $n = |V_t|$ is the number of nodes in the graph.

**Connecting Theory to Practice:** While a theoretical complexity of $\mathcal{O}(n^3)$ might seem prohibitive, it is crucial to note that our entire framework is explicitly designed to ensure $n$ remains small.

- **Bounded Graph Size:** As enforced by our **Algorithm 1**, which clusters nearby LiDAR readings, the number of nodes $n$ is not a function of the **observation dimension** (128) but of the **number of perceived entities.**

- **Empirical Validation:** The practical cost of this $\mathcal{O}(n^3)$ computation is trivial **because $n$ is small**. This is empirically validated by our runtime analysis in figures 20, 21 and 22 in the next section, where we note our method's total runtime is only **1.1x to 1.25x**, for per episode training run, that of vanilla PPO across all the benchmarks. This marginal increase confirms that the time spent on PPO's policy/value updates far exceeds the time spent on our graph and $\mathcal{R}_{\text{eff}}$ calculations. Additionally, this marginal increase in runtime is completely countered by the much faster convergence during training from figure 5.

In summary, while the formal complexity is $\mathcal{O}(n^3)$, our principled graph construction ensures $n$ is small and bounded, making the practical computational cost negligible and significantly lower than other SOTA baselines like GCPO.

## A.11 EFFICIENCY OF OUR METHOD

To further justify the practicality of our method, we also compute the per episode rollout time based on the process time (in order to provide a fair comparison by ignoring the sleep time, cache misses etc) and compare those across all methods. The plots in figures 20, 21 and 22 show the median process time per episode along with the percentiles as done previously. As expected, PPO runs the fastest since it does not incur any additional computation whereas PPO+Ent incurs a small additional cost for the entropy calculation, but the ratio of median runtime per episode for Our method vs PPO is only between $1.2 - 1.25$ on navigation environment, which further reduces to between $1.1 - 1.17$ on a more complex building environment. This is much better than baseline ratio of SRL-Std to PPO at $1.29 - 1.51$ on navigation and the best baseline GCPO's ratio to PPO at $1.38 - 1.62$. Notice that the graph construction and $\mathcal{R}_{\text{eff}}$ calculation of our method can be done on any standard CPU core within milliseconds per state observation, which further becomes favorable as the environment gets complex.

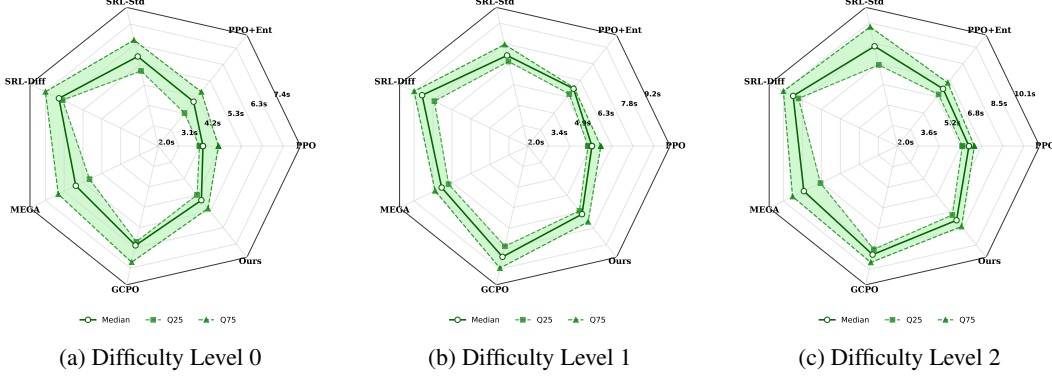

(a) Difficulty Level 0      (b) Difficulty Level 1      (c) Difficulty Level 2

Figure 20: Navigation Environment Times

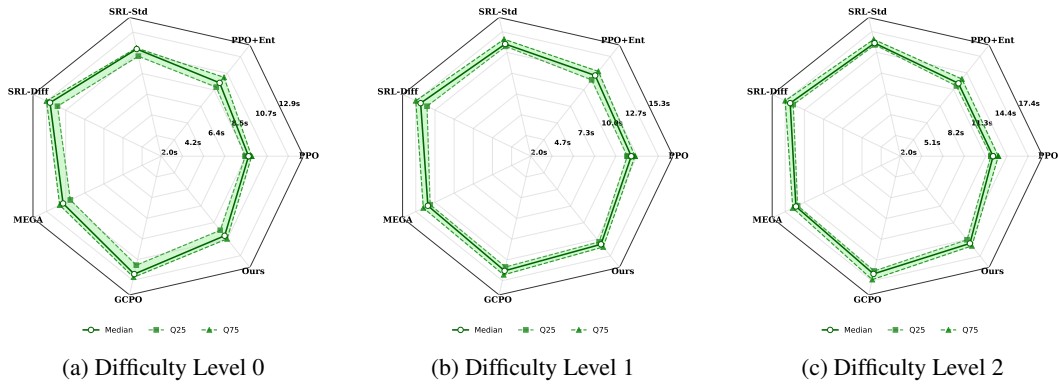

Figure 21: Building Environment Times

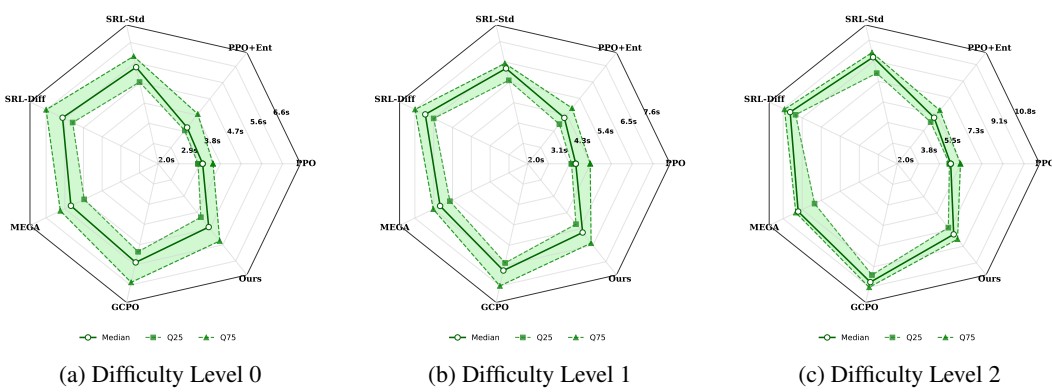

Figure 22: Fading Environment

## A.12 ABLATION FOR THE CENTRAL NODES CONNECTIVITY OF THE OBJECT TYPE SUBGRAPHS

Another key component of the graph construction algorithm 1 is the connectivity of central nodes based on the degree for each type $(V_{\mathcal{S}^i}, i \in [P])$. We provided a rationale for this in section A.5.1 and justify this empirically in figure 23. While the success rate does not fluctuate much since the agent still navigates to the goal eventually, we see a much more pronounced effect on the number of timesteps taken to reach the goal as well as the median reward the agent receives on its trajectories. For eg - in the navigation environment we see around 10% increase in the number of timesteps to reach the goal and around 14% increase for the building environment. Correspondingly, the agent's rewards are reduced by almost 8.4% and 12% respectively. The fading environment is relatively less sensitive to this component of the graph construction.

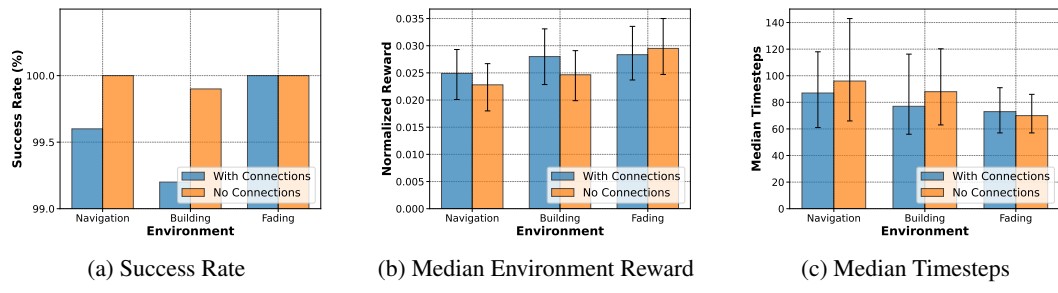

Figure 23: Success Rate, Median Timesteps and Median Normalized Reward across the environments for the ablation of algorithm 1

### A.13 EVOLUTION OF THE STANDARD DEVIATION $\sigma_{\mathcal{A}}(t)$ OF THE ACTION DISTRIBUTION DURING TRAINING

Figures 24, 25 and 26 compare the evolution of the standard deviation parameter $\sigma_{\mathcal{A}}$ over the course of training. Our primary emphasis here is to compare this against PPO (and PPO+Ent) while connecting it to the convergence results from sections 4 and experiments of 5.2. Since during training we sample the agent's action from the $\mathcal{N}(\mu_\theta(L_{obs}(t)), \sigma^2)$, there is a natural exploration component in this formulation, however *high deviation* values for longer can lead to - slower convergence, more timesteps to reach the goal and furthermore not reaching the goal in many cases due to excessive exploration in the environment. Thus, we argue that a higher negative slope of standard deviation curve, but not too extreme (as this will hinder the exploration in the environment leading to poor outcomes), is favorable. This rationale has been also thoroughly justified in the previous sections of this work.

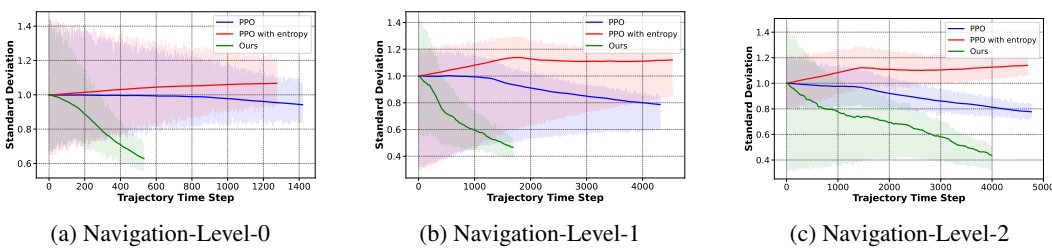

(a) Navigation-Level-0     (b) Navigation-Level-1     (c) Navigation-Level-2

Figure 24: Standard Deviation $\sigma_{\mathcal{A}}(t)$ of the action distribution during training in the Navigation environments.

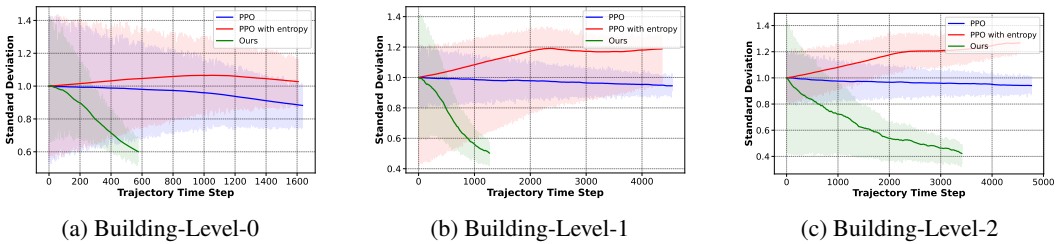

(a) Building-Level-0     (b) Building-Level-1     (c) Building-Level-2

Figure 25: Standard Deviation $\sigma_{\mathcal{A}}(t)$ of the action distribution during training in the Building environments.

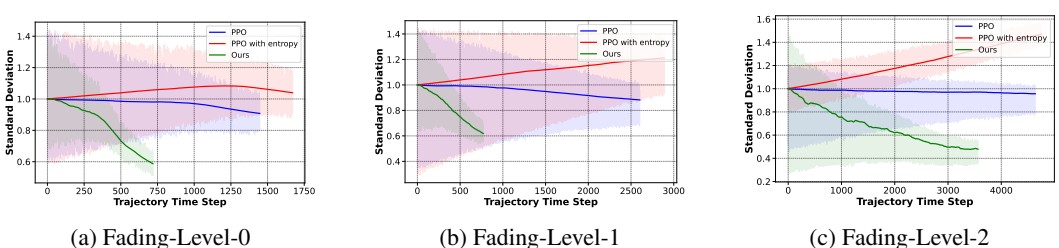

(a) Fading-Level-0     (b) Fading-Level-1     (c) Fading-Level-2

Figure 26: Standard Deviation $\sigma_{\mathcal{A}}(t)$ of the action distribution during training in the Fading environments.

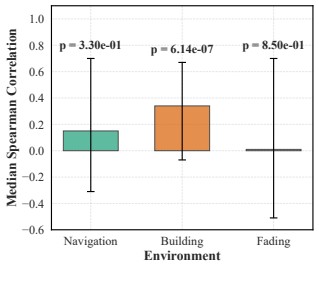

(a) Difficulty Level 2

Figure 27: Spearman Correlation coefficients between the value function and $-\mathcal{R}_{\text{eff}}$ over level 2 for the environments. We note the weaker correlation values are primarily due to the difficulty in approximating the value function as noted in several works (Moon et al., 2023; Engstrom et al., 2020) in conjunction with other factors affecting the training.

### A.14 Extension to Dense Environment Reward setup

We provide an extension of our experiments to the case where the reward obtained from the environment is dense, ie, $r_{ext}$ is used at every timestep $K = 1$ in contrast to the $K = 25$ mentioned in main results of section 5. We perform empirical validation across all the environments on both - the easier level-0 difficulty and the much more difficult level-2, which covers both ends of the spectrum. We have several observations from the results described in tables 6 and 7:

- Inline with the results from figure 6 (correlation between the value function and $-\mathcal{R}_{\text{eff}}$) and the theoretical results, the augmentation of $r_{int}$ keeps the results consistent, without interfering the learning and convergence process.

- For difficulty level 0, where the results of our sparse setting already reach complete success, the observed improvements are marginal.

- For difficulty level 2, which is a much more complex setup due to significantly larger number of obstacles and goal reaching difficulty (section A.2.2), we again note that $r_{int}$ does not interfere and rather serve as a reward shaping mechanism to assist the learning process.

Lastly, we also emphasize that the focus of our work is to improve the convergence of sparse goal conditioned online RL, and this set of experiments is to merely justify that breadth of applicability of our propsed $\mathcal{R}_{\text{eff}}$.

Table 6: The comparison of evaluation results, over difficulty level 0, for policies trained via the **Sparse** formulation where the standard environment reward is provided at every $K = 25$ steps (results copied from figures 2, 3 and 4 against the **Dense** formulation where we utilize the standard environment reward at every timestep $K = 1$. The columns for *Normalized Reward* and *Timesteps* detail the median results along with the $25^{th}$ and $75^{th}$ percentile results.

| Env | Setup | Success Rate ↑ | Normalized Reward ↑ | Timesteps ↓ |
|---|---|---|---|---|
| Navigation-level-0 | Sparse | 99.90% | 0.03091 (0.02709 ; 0.03545) | 55 (46 ; 69) |
| | Dense | 100.0% | 0.03124 (0.02771 ; 0.03633) | 51.5 (43 ; 67.75) |
| Building-level-0 | Sparse | 99.60% | 0.02833 (0.02533 ; 0.03250) | 60 (48 ; 78) |
| | Dense | 100.0% | 0.02974 (0.02702 ; 0.03413) | 56.25 (44 ; 77) |
| Fading-level-0 | Sparse | 99.90% | 0.03075 (0.02868 ; 0.03340) | 53 (44.75 ; 61) |
| | Dense | 100.0% | 0.03218 (0.02959 ; 0.03491) | 52 (42.5 ; 59.5) |

Table 7: The comparison of evaluation results, over difficulty level 2, for policies trained via the **Sparse** formulation where the standard environment reward is provided at every $K = 25$ steps (results copied from figures 2, 3 and 4 against the **Dense** formulation where we utilize the standard environment reward at every timestep $K = 1$. The columns for *Normalized Reward* and *Timesteps* detail the median results along with the $25^{th}$ and $75^{th}$ percentile results.

| Env | Setup | Success Rate ↑ | Normalized Reward ↑ | Timesteps ↓ |
|---|---|---|---|---|
| Navigation-level-2 | Sparse | 55.50% | 0.00682 (-0.00169 ; 0.00996) | 242 (97 ; 1000) |
| | Dense | 100.0% | 0.02790 (0.02238 ; 0.03465) | 88 (63 ; 122) |
| Building-level-2 | Sparse | 88.40% | 0.01339 (0.00994 ; 0.01679) | 168 (89 ; 518.75) |
| | Dense | 100.0% | 0.02420 (0.01959 ; 0.02908) | 98 (71 ; 137) |
| Fading-level-2 | Sparse | 61.10% | 0.00702 (0.00004 ; 0.01000) | 248 (85 ; 1000) |
| | Dense | 100.0% | 0.02790 (0.02228 ; 0.03380) | 92 (69 ; 114) |

## A.15 DISCUSSION ON GENERALIZATION BEYOND LiDAR

We emphasize that the core requirement of our framework is *not* LiDAR, but rather a state $s_t$ that can be meaningfully decomposed into a graph of interacting entities (e.g., agent, goal, obstacles and such). The LiDAR data used in our experiments is merely a practical *instantiation* of this, as it directly provides categorized, egocentric information that our Algorithm 1 can parse into nodes and edges. We have also characterized this experimental choice by attributing to the extensive success of the use of sensory data (such as LiDAR) in both - simulation and real world environments (as discussed in section A.2.2). The following extensions, which essentially cover almost real world implementations, support of our arguments:

- **Raw Pixel Inputs (Vision):** For environments where the state $s_t$ is an image, the graph can be constructed by deploying a perception frontend. For example, a pre-trained (or jointly trained) object detector or keypoint extractor identifies the "agent," "goal," and various "obstacles" in the scene. The bounding boxes or keypoints of these detected entities serve as the nodes $V_t$. Edges $E_t$ and weights $W_t$ are then be defined via the spatial (e.g., pixel distance) or semantic relationships, and the $\mathcal{R}_{\text{eff}}$ computation directly proceeds as discussed in our Algorithm 1

- **Tabular MDPs (e.g., Grid Worlds):** In a discrete grid world, the graph construction is even more straightforward. The **nodes** $V_t$ are the agent's current cell, the goal cell, and all cells identified as obstacles or hazards. The **edges** $E_t$ represent physical adjacency (e.g., connecting all non-obstacle cells that share a border), with **weights** $W_t$ set to unity. In this configuration, $\mathcal{R}_{\text{eff}}(\text{agent}, \text{goal})$ serve as a robust, non-Euclidean distance metric that accounts for all walls and pathways, providing a dense and informative reward signal.

- **General Sensory Inputs**: are trivial to handle with our current construction directly.

This flexibility demonstrates that our core contribution—using $\mathcal{R}_{\text{eff}}$ as a goal-directed structural reward—is a general principle that can be adapted to any MDP where the state can be abstracted into an entity-based graph.

## A.16 GENERALIZATION OF $\mathcal{R}_{\text{EFF}}$ TO OTHER ENVIRONMENTS

In the previous section we characterized the generality of our algorithm and proposed metric to various types of inputs, albeit within navigation and manipulation tasks. We start by noting that these two types of tasks describe a substantial percentage of the real world deployments of RL - humanoids, delivery bots, warehouse robots and such. We have a general implementation as follows:

- *Nodes:* by discretizing the state space into regions, eg using a VQ-VAE, where each codebook vector becomes a node in the graph $\mathcal{G}$

- *Edges and Weights:* connecting temporally adjacent or regions in vicinity, with the weights representing transition probabilities or inverse distance.

- *Computing $\mathcal{R}_{\text{eff}}$:* by considering two nodes from the graph $\mathcal{G}$, the $\mathcal{R}_{\text{eff}}$ can be calculated and minimized against a baseline set of $\mathcal{R}_{\text{eff}}$ values computed over some expert's data or simply using the highest reward states. We discuss this with the following two examples.

Example environments for generalization of $\mathcal{R}_{\text{eff}}$

- **Antmaze (OGBench (Park et al., 2025)**): The following simple construction can generate a graph over which we compute $\mathcal{R}_{\text{eff}}$

    1. *Nodes:* each cell or a group of cell describes a node in the graph $\mathcal{G}$. The cells are binned by their respective categories of - the agent location, the obstacle (wall) and the cells containing the goal.
    2. *Edges and Weights:* are computed simply by inverse distance of the cells on the grid to incorporate for basic geometry, or by connecting neighboring cells for a more trivial graph.
    3. $\mathcal{R}_{\text{eff}}$: the computation simply follows our eq 2 between the agent and goal node cells.

- **Locomotion tasks (e.g., Half-Cheetah) (Brockman et al., 2016)**:

    1. *Nodes:* The graph nodes $V$ are the agent's primary body parts (e.g., for Half-Cheetah, this includes the torso, thighs, shins, and feet). The agent's state $s_t$ (containing joint positions) defines the spatial embedding of these nodes at time $t$.
    2. *Edges & Weights:* The graph $\mathcal{G}_t$ could be fully connected, where the weight $w_{ij}(t)$ between any two nodes $i$ and $j$ is a function of their spatial distance, e.g., $w_{ij}(t) = 1/(||pos_i(t) - pos_j(t)||_2 + \epsilon)$. This graph's Laplacian, $L_t$, holistically represents the agent's *entire pose* at time $t$.
    3. $\mathcal{R}_{\text{eff}}$: The task (e.g., learning a stable gait) can be framed as matching a target pose. We can define a set $\hat{R}_{target}$ of "target effective resistance" $\mathcal{R}_{\text{target}}$ values (computed either from various expert's "stable" poses or high reward states/poses) between two key nodes (e.g., front_foot and back_foot).
    4. *Intrinsic Reward:* The intrinsic reward would be a shaping reward to match this target: $r_{int} = -(\mathcal{R}_{\text{eff}}(t) - \mathcal{R}_{\text{target}})^2$, where $\mathcal{R}_{\text{target}} \in \hat{R}_{target}$. This guides the agent to learn a stable pose configuration, demonstrating the flexibility of $\mathcal{R}_{\text{eff}}$ as a holistic state descriptor.

## A.17 COMPARISON WITH LEARNED METRIC APPROACHES

Recent advancements in Quasimetric Learning (QRL) (Wang et al., 2023; Liu et al., 2024) have demonstrated the utility of learning distance metrics from interaction data to capture the underlying geometry of the state space. While both QRL and our Effective Resistance ($\mathcal{R}_{eff}$) formulation serve as distance metrics to guide exploration, they differ fundamentally in their derivation, sample efficiency, and interpretability.

**Analytic vs. Learned Derivation:** Approaches like Wang et al. (2023) learn a metric function $d(s, s')$ via optimization over extensive transition data. While this allows the metric to adapt to complex, non-Euclidean dynamics, it inherently requires significant samples to converge to a useful representation. In contrast, our method analytically computes $\mathcal{R}_{eff}$ from the graph Laplacian at each timestep based on the local observation. This analytic nature eliminates the need for a metric-learning phase, allowing our intrinsic reward to provide a dense, meaningful signal immediately upon interaction.

**Inductive Bias and Interpretability:** A key advantage of our approach is its strong inductive bias: we posit that the "difficulty" of reaching a goal is structurally correlated with the resistive connectivity of the graph constructed from the state. This yields high interpretability. As illustrated in Figure 1, a reduction in $\mathcal{R}_{eff}$ directly corresponds to tangible structural improvements, such as the widening of a bottleneck or the opening of a new pathway. Conversely, metrics learned via function approximation can act as "black boxes," making it difficult to ascertain whether the learned distance reflects true reachability or spurious correlations in the training data.

**Theoretical Guarantees:** Finally, because our metric is grounded in spectral graph theory, we can derive formal guarantees regarding connectivity preservation and robust navigation (as detailed in Lemma 1 and Theorem 1). Establishing similar theoretical bounds for metrics learned purely from stochastic environment interactions remains a challenging open problem.

## A.18 USE OF LLMS

We have used LLMs, particularly GPT and Perplexity, for paraphrasing some sentences to make them academically sound, basic proof reading and finding relevant literature works to ensure completeness and coverage.

