# OpenReview forum: "Graph-Theoretic Intrinsic Reward: Guiding RL with Effective Resistance"
_ICLR.cc/2026/Conference — ICLR 2026 Poster_

### Official Review · Reviewer_nmWg · 2025-10-22

**Soundness:** 2
**Presentation:** 1
**Contribution:** 2
**Rating:** 2
**Confidence:** 4

**Summary:**

This paper proposes a new metric termed effective resistance that can be used as an intrinsic motivation reward for goal-conditioned RL tasks. The paper hypothesizes that this reward is better than using one proportional to Euclidean distance to the goal. The paper also performs some analysis to show that the proposed intrinsic reward is almost unbiased with respect to the extrinsic reward, and that using the proposed intrinsic reward leads to improved sample complexity.

The empirical evaluation on a suite of environments called Safety-Gymnasium seek to show that the above proposed theoretical guarantees hold, and that the proposed technique outperforms some reasonable baselines.

**Strengths:**

Overall the idea is somewhat novel and might be a useful addition to the literature.
* The proposed technique is interesting. Using ideas that consider graph flows to estimate how easy it is to navigate from one point to another has been seen to be useful in the past [1], and this modern revival of the technique could present some benefits.
* Dynamic graph updates are very useful, allowing an environment that changes over time.
* Theoretical analysis seems to be sound and gives some confidence that the proposed approach will not converge to a suboptimal solution and will help with some local exploration.
* The evaluation methodology seems robust and statistically sound, and I especially appreciate the 1000 episode evaluations (5 training seeds and 200 episode evals per seed), which should capture variance in the policy and variance in the training.
* The baselines that are used in the evaluation seem mostly reasonable. There are caveats here that I expand on in weaknesses.



## References
[1] Şimşek, Ö., Wolfe, A.P. and Barto, A.G., 2005, August. Identifying useful subgoals in reinforcement learning by local graph partitioning. In Proceedings of the 22nd international conference on Machine learning (pp. 816-823).

**Weaknesses:**

There are some issues that keep this paper from being of a quality that I can confidently recommend for acceptance:
* This paper is trying to suggest a new intrinsic motivation based on effective resistance in a graph. But seems to be specific to robotics type problems with objects present in the environment causing navigational or manipulational difficulties. If specific to robotics problems, the setup and writing should clarify and try to position itself accordingly so that it will attract the same community of researchers. It also does not compare to other GCRL methods for exploration in literature [1, 2, 3], or more up to date GCRL benchmarks like OGBench [4].
* Part of the issue here is that the problem setup is specific to continuous state spaces and a 2 dimensional action space (Section 3.1).
* My understanding is that the intrinsic reward is only calculated when the reward enters the agent's field of view. This seems like it will help mostly with local exploration instead of more generally.
* The baselines proposed do not seem to be exploiting the structure of GCRL based problems from what I can tell. One of [1] or [3] would be great additions to show how effective the proposed intrinsic reward is in GCRL problems specifically.


## References
[1] Grace Liu, Michael Tang, and Benjamin Eysenbach. A single goal is all you need: Skills and exploration emerge from contrastive RL without rewards, demonstrations, or subgoals. In The Thirteenth International Conference on Learning Representations, 2025. URL https: //openreview.net/forum?id=xCkgX4Xfu0

[2] Ma, Y.J., Yan, J., Jayaraman, D. and Bastani, O., 2022. How Far I'll Go: Offline Goal-Conditioned Reinforcement Learning via $ f $-Advantage Regression. arXiv preprint arXiv:2206.03023.

[3] Durugkar, I., Tec, M., Niekum, S. and Stone, P., 2021. Adversarial intrinsic motivation for reinforcement learning. Advances in Neural Information Processing Systems, 34, pp.8622-8636.

[4] Park, S., Frans, K., Eysenbach, B. and Levine, S., 2024. Ogbench: Benchmarking offline goal-conditioned rl. arXiv preprint arXiv:2410.20092.

**Questions:**

* Could the authors clarify how more general exploration will be handled under the given scheme? Perhaps contrast with [3] from the weakness section, since that is an approach that handles more general exploration?
* Could approaches like Quasimetric learning [1] also learn some metric like effective resistance?

## References
[1] Wang, T., Torralba, A., Isola, P. and Zhang, A., 2023, July. Optimal goal-reaching reinforcement learning via quasimetric learning. In International Conference on Machine Learning (pp. 36411-36430). PMLR.

---

> ### Author Response · Authors · 2025-11-19
> **Response to Reviewer nmWg (1/3)**
>
> We thank Reviewer nmWg for their time and for providing a detailed, critical review. The review highlights perceived issues with generality and baseline comparisons. We are confident that by clarifying these points and proposing specific additions, we can demonstrate the novelty and significance of our contribution. We have addressed all concerns regarding baseline comparisons, method generality, and the purpose of our reward, including implementing the reviewer's requested baseline AIM. We have added a dedicated analysis section and updated all main results accordingly. We invite the reviewer to engage in further discussion should any points require additional clarification and humbly ask you to reconsider your evaluation in light of the extensive revisions and strengthened contribution of our work.
>
> ## Weakness 1: Domain Specificity & GCRL Baselines
> ### Reviewer's Comment
> "This paper is trying to suggest a new intrinsic motivation based on effective resistance in a graph. But seems to be specific to robotics type problems with objects present in the environment causing navigational or manipulational difficulties. If specific to robotics problems, the setup and writing should clarify and try to position itself accordingly so that it will attract the same community of researchers. It also does not compare to other GCRL methods for exploration in literature [1, 2, 3], or more up to date GCRL benchmarks like OGBench [4]."
>
> ### Our Response:
>
> We respectfully disagree that our method is domain-specific. We chose robotics (Safety-Gym, Section 5) as a testbed because its sparse rewards and dynamic obstacles are notoriously difficult. However, the core requirement of our framework is simply a state $s_{t}$ that can be decomposed into interacting entities. **The LiDAR vector (Appendix A.2.2)** is just one practical instantiation.
>
> To explicitly demonstrate this generality, we have added **Appendix A.15 (Page 41) and Appendix A.16 (Page 42)**. These sections detail how our framework extends to other modalities, such as using **perception frontends for vision-based tasks (lines 2187-2199) or defining nodes and edges for tabular grid worlds and complex locomotion tasks like Half-Cheetah (lines 2230-2247)**.
>
> Regarding the baselines, we emphasize that our method is on-policy. **Several suggested references ([2] Ma et al., [4] OGBench) are explicitly for offline RL, making the direct comparison difficult.** On the other hand, we did compare against **GCPO (Section 5, Page 6)**, a SOTA on-policy GCRL method. In order to address the reviewer's specific request, we have now integrated **AIM [3] and NGU as additional baselines across all 9 environments. As shown in the updated Figures 2, 3, and 4 (Pages 7-8),** our method significantly outperforms both, **with AIM achieving only less than a third of our success rate on the highest difficulty levels of the building and navigation tasks, with similar observations on other metrics as well.** .
>
> ## Weakness 2: State and Action Space
> ### Reviewer's Comment
> "Part of the issue here is that the problem setup is specific to continuous state spaces and a 2 dimensional action space (Section 3.1)."
> ## Our Response
> We clarify that these characteristics—continuous states and 2D actions—are simply  benchmark specifications and completely disentangled from our proposed algorithm.
>
> State Space: Our method is expressly designed to handle complex, high-dimensional continuous inputs. **As detailed in Appendix A.2.2 (Page 23)**, it operates effectively on sensor data vectors as large as 128-dimensions. This capability is a core strength, demonstrating that our graph-based reward scales to rich sensory data.
>
> Action Space: The 2D action space (throttle and steering) is a property of the "Point" agent in the benchmark **(Section 3.1, Page 3)**. Our intrinsic reward formulation **(Eq. 1, 3)** is entirely algorithm-agnostic. It modifies the reward signal $r_{total} = r_{ext} + \alpha r_{int}$, placing absolutely no constraints on the action space dimensionality. It is compatible with discrete, continuous, or hybrid action spaces.
>
> We note that these environment choices align with standard practices in recent literature, including accepted ICLR works such as [6], ensuring our evaluation is comparable and rigorous.

---

> ### Author Response · Authors · 2025-11-19
> **Response to Reviewer nmWg (2/3)**
>
> ## Weakness 3: Reward Calculation and Local Exploration
> ### Reviewer's Comment
> "My understanding is that the intrinsic reward is only calculated when the reward enters the agent's field of view. This seems like it will help mostly with local exploration instead of more generally."
> ## Our Response
> This is a sharp observation, and we want to emphasize that this behavior is a **deliberate design choice, not a limitation.**
>
> As shown in **Equation 3 (Page 4)**, our intrinsic reward $r_{int}$ is non-zero only when the goal is visible. This is because our method is explicitly designed as a goal-directed planning signal, not a general exploration bonus like RND or [3], which aim to explore the entire state space and can be "distracted" by irrelevant information.
>
> Instead, our method answers the specific question: "Given that I can see the goal, what is the most robust path to it?" Our $\mathcal{R}_{\text{eff}}$-based reward provides a dense signal for this local navigation phase, guiding the agent around obstacles **(as illustrated in Figure 1, Page 2)** in ways simple metrics cannot.
>
> The complementary task of global exploration (finding the goal initially) is handled by the base PPO algorithm's entropy bonus and is powerfully encouraged by the large $+\beta$ reward for initial goal acquisition **(Equation 3)**. This separation of concerns—using $r_{int}$ for robust local planning and the base algorithm for global search—is a key feature of our approach.
>
> We additionally highlight that local / egocentric reward is a more practical choice. For majority of the environments, the MDP are *Partially Observable* and thus we cannot expect to incorporate the entire state in the reward, or more generally, the model dynamics.
>
>
> ## Weakness 4: GCRL Baselines
> ### Reviewer's Comment
> "The baselines proposed do not seem to be exploiting the structure of GCRL based problems from what I can tell. One of [1] or [3] would be great additions to show how effective the proposed intrinsic reward is in GCRL problems specifically."
>
> ### Our Response
>
> We thank the reviewer for this specific and constructive suggestion. We agree that comparing against a GCRL-specific intrinsic motivation method is crucial.
>
> **We have now implemented and evaluated AIM (Adversarial Intrinsic Motivation)** [3] across all our benchmarks. This baseline is directly relevant as it generates an intrinsic reward based on the Wasserstein distance between the policy's state visitation and a target distribution, specifically for goal-directed tasks.
>
> The results, now incorporated into **Figures 2, 3, and 4 (Pages 7-8)**, show that our method significantly outperforms AIM. For example, in the challenging Navigation-Level-2 task, our method achieves a 55.5% success rate compared to AIM's 18.5%. Similar trends hold for sample efficiency and convergence speed. We have updated the **Baselines subsection in Section 5 (Page 6) to include AIM and expanded our analysis to discuss these comparative results.**
>
>
> ## Question 1: General Exploration and Contrast with [3]
> ### Reviewer's Comment
> "Could the authors clarify how more general exploration will be handled under the given scheme? Perhaps contrast with [3] from the weakness section, since that is an approach that handles more general exploration?"
>
> ## Our Response
>
> This is an excellent question, directly linking to Weakness 3.
>
> **General Exploration:** As stated in our response to W3, general exploration (finding the goal) is handled by the base RL algorithm (PPO, which includes stochasticity and an entropy bonus). Our method acts as a complementary reward that creates an "express lane" to the goal once it is found. We have justified this via both theoretical convergence guarantees (Lemma 4) and empirical validation of faster loss plateaus (Figure 5) alongside superior performance.
>
> **Contrast with [3]:** [3] (Durugkar et al., 2021) is a general exploration method. It uses an adversary to push the agent to visit novel, hard-to-reach states, regardless of a goal. Our method is a goal-directed method. It uses spectral graph theory to find the most robust path to a known goal. We note that these two methods solve different and complementary problems. However, as suggested by the reviewer, for completeness, we have updated our writeup and experimental section to include AIM [3] as a relevant baseline. As shown in the updated Figure 2, our goal-directed approach significantly outperforms AIM's general exploration strategy on these specific tasks.

---

> ### Author Response · Authors · 2025-11-19
> **Response to Reviewer nmWg (3/3)**
>
> ## Question 2: Quasimetric Learning [1] and Effective Resistance
> ### Reviewer's Comment
> "Could approaches like Quasimetric learning also learn some metric like effective resistance?
> ### Our Response
> This is a very insightful question that touches on the distinction between learned and analytic metrics. It is certainly plausible that a quasimetric approach [1], which learns a distance function from data, could approximate a metric with properties similar to effective resistance ($\mathcal{R}_{\text{eff}}$). Both serve as distance metrics that can capture the underlying geometry of the state space. However, the crucial difference lies in how the metric is obtained:
>
> **Learned vs. Analytic:** [1] learns its metric from interaction data, requiring significant samples to converge to a useful representation. In contrast, our method analytically computes $\mathcal{R}_{\text{eff}}$ from the state's (locally observable) structure (the graph) at each timestep.
>
> **Guarantees & Interpretability:** Because our metric is computed directly from the graph Laplacian, it is (a) highly interpretable (as shown in Figure 1), (b) grounded in spectral graph theory, and (c) comes with theoretical guarantees (Lemma 1) that it represents "structural accessibility." A learned metric would lack these inherent properties, would likely be less sample-efficient to obtain, and would offer no guarantees of what it has actually learned (e.g., it might overfit to spurious correlations).
>
> We thank the reviewer for this connection.
>
> **References**
>
> [1] Grace Liu, Michael Tang, and Benjamin Eysenbach. A single goal is all you need: Skills and exploration emerge from contrastive RL without rewards, demonstrations, or subgoals. In The Thirteenth International Conference on Learning Representations, 2025. URL https: //openreview.net/forum?id=xCkgX4Xfu0
>
> [2] Ma, Y.J., Yan, J., Jayaraman, D. and Bastani, O., 2022. How Far I'll Go: Offline Goal-Conditioned Reinforcement Learning via
> -Advantage Regression. arXiv preprint arXiv:2206.03023.
>
> [3] Durugkar, I., Tec, M., Niekum, S. and Stone, P., 2021. Adversarial intrinsic motivation for reinforcement learning. Advances in Neural Information Processing Systems, 34, pp.8622-8636.
>
> [4] Park, S., Frans, K., Eysenbach, B. and Levine, S., 2024. Ogbench: Benchmarking offline goal-conditioned rl. arXiv preprint arXiv:2410.20092.
>
> [5] Badia, A. P., Sprechmann, P., Vitvitskyi, A., Guo, D., Piot, B., Kapturowski, S., Tieleman, O., Arjovsky, M., Pritzel, A., Bolt, A., and Blundell, C. “Never Give Up: Learning Directed Exploration Strategies,” 2020.
>
> [6] Zubia, M., Simão, T. D., and Jansen, N. “Robust Transfer of Safety-Constrained Reinforcement Learning Agents,” ICLR 2025.
>
> [7] Burda, Y., Edwards, H., Storkey, A., and Klimov, O. “Exploration by Random Network Distillation,” 2018.
>
> [8] Wang, T., Torralba, A., Isola, P. and Zhang, A., 2023, July. Optimal goal-reaching reinforcement learning via quasimetric learning. In International Conference on Machine Learning (pp. 36411-36430). PMLR.

---

> > ### Comment · Reviewer_nmWg · 2025-11-27
> > **Response to Author Rebuttal**
> >
> > I thank the authors for the detailed response to all the reviews and for updating the paper with suggestions that were made.
> >
> > I have no issues with the clarifications that the authors have made. Responding to some specific points below:
> > > We clarify that these characteristics—continuous states and 2D actions—are simply benchmark specifications and completely disentangled from our proposed algorithm.
> >
> > I am happy with the addition to the appendix. My intention with bringing up the specificity of the state and action space setup is so that the writing can be modified to show the generality of the approach upfront, instead of via clarifications in the appendix. I am satisfied with the current state of the paper regarding the problem setup, but I suggest the authors consider writing the MDP formulation in set notation just for generality. However it is a stylistic suggestion and not a requirement.
> >
> > ## Comparison between Quasimetric Learning and Effective Resistance
> >
> > I liked this contrast. I encourage the authors to include a succinct version of this discussion in the main paper. The contrast between a method that can learn arbitrary distance metrics based on dynamics and one that has some inductive bias that allows interpretability would be useful for potential readers.
> >
> > Overall I am satisfied with the author response. I will be raising my score. But I will consider the other reviewers discussion and opinions before I settle on an exact score.

---

> ### Author Response · Authors · 2025-11-28
> **Replying to reviewer regarding Comparison between Quasimetric Learning and Effective Resistance**
>
> Thank you for your continued engagement and your positive reception of our rebuttal. We hope you are doing well. We completely agree with your suggestion: explicitly contrasting our method with Quasimetric Learning helps clarify the trade-off between flexible, learned metrics and those with strong inductive biases like ours.
>
> Per your recommendation, we have updated the manuscript to include this discussion:
>
> 1. Main Paper **(Section 3.2, Lines 194–199)**: We added a succinct paragraph highlighting the distinction between learned metrics (which require extensive data) and our analytic formulation (which offers immediate interpretability and structural guarantees).
>
> 2. **Appendix A.17**: We included a comprehensive section detailed in our previous response, elaborating on the differences in derivation, sample efficiency, and theoretical grounding.
>
> We believe these additions significantly improve the paper's clarity and positioning within the broader literature. We hope that these revisions, combined with our previous responses, fully address your comments, and we would be grateful if you could consider reflecting these improvements in your final assessment of our work.

---

### Official Review · Reviewer_Pwps · 2025-11-03

**Soundness:** 3
**Presentation:** 3
**Contribution:** 3
**Rating:** 4
**Confidence:** 2

**Summary:**

This paper introduces a novel intrinsic reward mechanism for reinforcement learning in sparse reward environments based on effective resistance from spectral graph theory. The key idea is to construct a time-evolving graph from the agent's observations (specifically LiDAR data) where nodes represent the agent, goal, and environmental objects, and edges encode proximity relationships. The intrinsic reward is defined as the negative change in effective resistance between the agent and goal nodes, encouraging the agent to seek configurations that improve structural accessibility to the goal.

**Strengths:**

1. The application of effective resistance from spectral graph theory to RL is creative and theoretically grounded.
2. The paper provides a comprehensive theoretical analysis with multiple lemmas and a main theorem.

**Weaknesses:**

1. The method is specifically designed for environments where meaningful graph construction from observations is possible. The reliance on LiDAR data limits applicability to certain domains, and it's unclear how this would extend to other observation modalities or higher-dimensional state spaces.
2. Algorithm 1 involves many design choices (clustering threshold τ, connectivity patterns, central node selection) that appear to require careful tuning. The sensitivity analysis (Section A.9) shows some robustness to τ, but the overall complexity raises concerns about generalizability.
3. While the paper compares against several baselines, most are relatively older methods. More recent state-of-the-art intrinsic motivation methods could strengthen the comparison.
4. While Section A.10 provides some runtime analysis, the computational cost of repeated graph construction and effective resistance computation could be prohibitive in real-time applications or larger graphs.

**Questions:**

1. How does the method scale to environments with many more objects or higher-dimensional observation spaces? What is the computational complexity as a function of graph size?
2. How does the method scale to environments with many more objects or higher-dimensional observation spaces? What is the computational complexity as a function of graph size?
3. How does the method perform in environments with dense rewards? Does the intrinsic reward provide benefits or potentially interfere with learning in such settings?
4. How does the method perform in environments with dense rewards? Does the intrinsic reward provide benefits or potentially interfere with learning in such settings?
5. Beyond τ, how sensitive is the method to other hyperparameters like α and β? The theoretical guidelines (Corollary 1) provide bounds, but practical selection seems to require empirical validation.
6. How does this approach compare to more recent intrinsic motivation methods like NGU, RND, or ICM on the same environments?

---

> ### Author Response · Authors · 2025-11-19
> **Response to Reviewer Pwps (1/3)**
>
> We sincerely thank Reviewer Pwps for their detailed, constructive, and insightful review. We appreciate the clear, actionable feedback, which has helped us identify several areas where we can strengthen and clarify our paper. We agree with many of the reviewer's points and have incorporated changes to address them. Our responses to the specific weaknesses and questions are below.
> We invite the reviewer to engage in further discussion should any points require additional clarification and respectfully ask you to consider increasing your score to reflect the substantial revisions and strengthened contribution of our work.
>
> ## Response to Weakness 1:Domain Specificity
> ### Reviewer's Comment:
> "The method is specifically designed for environments where meaningful graph construction from observations is possible. The reliance on LiDAR data limits applicability to certain domains, and it's unclear how this would extend to other observation modalities or higher-dimensional state spaces."
> ### Our Response:
> We thank the reviewer for this crucial point. We wish to clarify that our framework's core requirement is not LiDAR, but a state $s_t$ that can be decomposed into meaningful entities. LiDAR, as detailed in **Appendix A.2.2 (Page 23, lines 1197-1209)**, is a practical sensor in our robotics testbed that directly provides this entity-based state, reflecting its common use in real-world systems (e.g., Waymo) and benchmarks(CALVIN[1], LIBERO[2]).
>
> To explicitly address the reviewer's concern about generalization, we have added **Appendix A.15 (Page 41) and Appendix A.16 (Page 42)**. These new sections demonstrate the framework's flexibility, providing concrete formulations for other modalities. This includes using a perception frontend for raw pixel inputs **(App. A.15, lines 2187-2199)**, adapting to tabular grid worlds **(App. A.15, lines 2200-2207)**, and even modeling locomotion tasks like Half-Cheetah by using body parts as nodes **(App. A.16, lines 2230-2247)**. These additions show that our graph-based reward is a general principle, with LiDAR being just one practical instantiation.
>
> ### Weakness 2: Hyperparameter Tuning and Complexity (Algorithm 1)
> ### Reviewer's Comment
> "Algorithm 1 involves many design choices (clustering threshold $\tau$, connectivity patterns, central node selection) that appear to require careful tuning. The sensitivity analysis (Section A.9) shows some robustness to $\tau$, but the overall complexity raises concerns about generalizability."
> ### Our Response
> We appreciate this important concern. We have taken care to show that our design choices are not arbitrary but are in fact principled and empirically robust. We refer to the detailed justifications provided in the appendix.
>
> The complete rationale for each step of our graph construction **(Algorithm 1, Page 26)** is detailed in **Appendix A.5.1 (Page 28, lines 1472-1538)**. More importantly, the key hyperparameter $\tau$ is not selected by a simple grid search. We first provide a full analytical derivation for its valid range in **Appendix A.5.3 (Page 30, lines 1590-1695)**, grounding it in the environment's dynamics. We then empirically validate this in **Appendix A.9 (Page 36, Figure 19)**, where our sensitivity analysis shows that performance is stable across this analytically-derived range, confirming our choice is robust and not the result of "careful tuning."
>
> To ensure this was not missed, we have also, as the reviewer suggested, added a new sentence to the main text in **Section 3.2 (Page 4, lines 171-175)** that explicitly directs the reader to these appendices for the detailed rationale, analytical derivation, and empirical validation of our graph construction.
>
> We re-emphasize that this approach is much more principled than most of the works in the literature and simultaneously provide a higher degree of interpretability.

---

> ### Author Response · Authors · 2025-11-19
> **Response to Reviewer Pwps (2/3)**
>
> ## Weakness 3: Older Baselines
> ### Reviewer's Comment
> "While the paper compares against several baselines, most are relatively older methods. More recent state-of-the-art intrinsic motivation methods could strengthen the comparison."
>
> ### Our Response
>
> We thank the reviewer for this fair and valuable suggestion. We respectfully note that our original submission did include **GCPO, a 2024 SOTA on-policy baseline (Section 5, lines 303-308)** as well as **MEGA**. However, to thoroughly address this concern and strengthen our evaluation, we have now fully integrated two additional SOTA baselines, NGU and AIM, as requested.
>
> We have added these methods to our Baselines discussion in **Section 5 (Page 6, lines 296-302).** More importantly, we have re-run our entire empirical study across all 9 environments. All main results plots—**Figures 2, 3, and 4 (Pages 7-8)**—have been updated to include a direct comparison against NGU and AIM. These new results demonstrate that our method's significant performance gains hold, and often widen, against these strong, modern methods. For instance, the success rates are over **3x** and **4x** for the highest difficulty level for navigation and building environments respectively. Similar observations hold for the other metrics.
>
>
> ## Weakness 4: Computational Cost
> ### Reviewer's Comment
> "While Section A.10 provides some runtime analysis, the computational cost of repeated graph construction and effective resistance computation could be prohibitive in real-time applications or larger graphs."
> ### Our Response
> This is a critical, practical concern, and we confirm that the computational cost is not prohibitive. To address this directly, we have added a formal complexity analysis in the revised paper in **Appendix A.10 (Page 36, lines 1937-1979).** This section details that while the theoretical bottleneck for computing $\mathcal{R}_{\text{eff}}$ is $\mathcal{O}(n^3)$ **(line 1948),** our graph construction algorithm (Algorithm 1) is expressly designed to keep the graph size $n$ small, making this computation trivial in practice.
>
> This theoretical efficiency is validated by our full empirical runtime analysis in **Appendix A.11 (Page 37, lines 1980-1992).** As shown in **Figures 20-22**, our method incurs only a marginal 1.1x-1.17x per-episode runtime overhead compared to vanilla PPO on complex tasks. This is significantly faster than other baselines like GCPO (up to 1.62x).
>
> As we now also highlight in the main text in **Section 5.1 (Page 8, lines 426-431)**, this negligible computational cost is overwhelmingly compensated for by the benefits it provides: faster training convergence **(shown in Figure 5)** and dramatically improved sample efficiency **(shown in Figure 4).**
>
> ## Question 1: Scaling and Computational Complexity
> ### Reviewer's Comment
> "How does the method scale to environments with many more objects or higher-dimensional observation spaces? What is the computational complexity as a function of graph size?"
> ### Our Response
> This question links to **Weaknesses 1 and 4**. We emphasize that scaling to high-dimensional observations (e.g., pixels) is managed by a perception frontend. The scaling of our method depends on the number of entities ($n = |V_t|$), not the observation dimension, as our **graph construction (Algorithm 1)** is designed to keep $n$ small.
>
> Regarding computational complexity, the bottleneck is the effective resistance computation ($\mathcal{O}(n^3)$ due to the pseudoinverse of the $n \times n$ Laplacian). However, since $n$ is bounded and small by construction in most practical environments, this computation remains very fast in practice. To directly address this, we have added a formal analysis in **Appendix A.10 (Page 36, lines 1937-1979)** explicitly stating the $\mathcal{O}(n^3)$ complexity and connecting it to our empirical results. We have provided the details for this in the weakness 4 above and refer to the corresponding sections of the paper draft as well.

---

> ### Author Response · Authors · 2025-11-19
> **Response to Reviewer Pwps (3/3)**
>
> ## Question 2: Performance in Dense Reward Settings
> ### Reviewer's Comment
> "How does the method perform in environments with dense rewards? Does the intrinsic reward provide benefits or potentially interfere with learning in such settings?"
> ### Our Response
> This is an excellent question. Our work is motivated by sparse rewards, but our method is also highly effective in dense reward settings, where the intrinsic reward $r_{int}$ acts as a beneficial reward shaping term. To demonstrate this, we have added **Appendix A.14 (Pages 40-41) with a new set of experiments where the extrinsic reward is provided at every timestep ($K=1$) instead of sparsely ($K=25$)**.
>
> Our results in Tables 6 and 7 (Pages 40-41) show that supplementing the dense extrinsic reward with our proposed $r_{int}$ leads to consistent improvements across all environments, particularly in the most difficult ones (e.g., Navigation-Level-2), with no interference in learning. This aligns with our theoretical analysis in Lemma 4 (Page 5), which suggests that $r_{int}$ acts as a variance-reducing baseline, stabilizing and accelerating learning even when dense rewards are available.
>
> ## Question 3: Sensitivity to $\alpha$ and $\beta$
> ### Reviewer's Comment
> "Beyond $\tau$, how sensitive is the method to other hyperparameters like $\alpha$ and $\beta$? The theoretical guidelines (Corollary 1) provide bounds, but practical selection seems to require empirical validation."
> ### Our Response
> We thank the reviewer for this question. We have taken great care to ensure that our hyperparameter selection is principled and not the result of per-task tuning. As we detail in **Appendix A.6 (Page 32, Table 3)**, our choices for $\beta$ and $\alpha$ are grounded in the **theoretical bounds from Corollary 1 (Page 5)**. Specifically, $\beta=5$ was chosen to dominate the maximum extrinsic reward (ensuring goal visibility is prioritized), and $\alpha=10$ was derived based on the ratio of extrinsic rewards to connectivity changes ($\delta_{max}$).
>
> Crucially, we emphasize that these single values were used consistently across all 9 environments (3 tasks $\times$ 3 difficulty levels) without any environment-specific tuning. This stands in stark contrast to methods requiring intensive hyperparameter sweeps. To highlight this robustness, we have added a new paragraph to **Appendix A.6 (Page 32, lines 1728-1730)** explicitly stating that these values were fixed globally, demonstrating the method's stability and generality.
>
> ## Question 4: Comparison to NGU, RND, or ICM
> ### Reviewer's Comment
> "How does this approach compare to more recent intrinsic motivation methods like NGU, RND, or ICM on the same environments?"
>
> ### Our Response
>
> As we discussed in our response to **Weakness 3**, this is a critical point. To provide a complete empirical comparison, we have added NGU (Never Give Up) and AIM (Adversarial Intrinsic Motivation) as baselines, as they represent the state-of-the-art in curiosity and GCRL exploration respectively. We have updated all **main result plots (Figures 2, 3, and 4 on Pages 7-8)** to include these methods across all 9 environments. As shown, our method consistently outperforms both, achieving higher success rates and faster convergence, demonstrating the specific value of our structural, goal-directed reward signal in these complex navigation tasks.
>
> **References**
> [1] Mees, O., Hermann, L., Rosete-Beas, E., Burgard, W. “CALVIN: A Benchmark for Language-Conditioned Policy Learning for Long-Horizon Robot Manipulation Tasks,” 2022.
>
> [2] Liu, B., Zhu, Y., Gao, C., Feng, Y., Liu, Q., Zhu, Y.,  Stone, P. “LIBERO: Benchmarking Knowledge Transfer for Lifelong Robot Learning,” 2023.

---

> ### Author Response · Authors · 2025-11-28
> **Requesting Response to Our Rebuttal**
>
> We hope the reviewer is doing well!
> We would like to bring the reviewer's attention to the deadline of the rebuttal period, and we would thus like to request the reviewer to increase their score if their questions and comments have been addressed.

---

### Official Review · Reviewer_FfQG · 2025-11-06

**Soundness:** 3
**Presentation:** 3
**Contribution:** 3
**Rating:** 8
**Confidence:** 2

**Summary:**

This paper proposes a reward shaping methods orignated from spectral graph theory to tackle reward sparsity in reinforcement learning setting. By modifying its instrinsic reward, the agent needs to maintain a graph of its surrounding environment first through its sensors (e.g. LIDARs) every timestep, then calculates its effective resistance between itself and the goal, which will be used as part of its reward construction.  They provide theoretical guarantees to prove they can learn a robust policy and also achieves faster convergence. Through experiments, they show their methods can beat state of the art baselines.

**Strengths:**

- Good originality: abstract objects into nodes in graph, and design rewards based on the constructed graph. An novel reward shaping methods that encourage the agent to get closer to goal.
- Quality: theoretically sound. Assumptions setup with proper citation and completely. Proved decreasing the effective resistance can also maintain connectivity on the graph. This paper also show the advantage of its algorithms emprically with extensive experiments.

**Weaknesses:**

- This methods can only be applied to specific domains, for example, robotics navigation tasks in the paper, in which the robots have a suite of sensors that are assumed not having noise and the robots having a good localization capability and can categorizes or recognize objects as nodes in the map.
- I would appreciate more explanation on how the graph is constructed and what reducing effective resistance would bring on the main text, but overall the paper is easy to follow.

**Questions:**

- To increase the impact of this paper, can this method apply to a more general MDP setting, e.g. continuous MDP or tabular MDP, how would you construct the graph in such MDPs? specifically, what are nodes/edges/weights in these settings?

---

> ### Author Response · Authors · 2025-11-19
> **Response to Reviewer FfQG (1/2)**
>
> We are incredibly encouraged by the "8: accept, good paper" rating and appreciate the constructive comments, which have helped us identify areas for clarification. We believe the substantial revisions made to the manuscript, especially concerning generalization, fully address your initial queries. We invite you to engage in further conversation should any points require additional discussion and respectfully request you consider raising your score to reflect the strengthened contribution of our work.
>
> ## Response to Weakness 1: Domain Specificity and Assumptions
> ### Reviewer's Comment
> "This methods can only be applied to specific domains, for example, robotics navigation tasks in the paper, in which the robots have a suite of sensors that are assumed not having noise and the robots having a good localization capability and can categorizes or recognize objects as nodes in the map."
>
> ### Our Response
> We thank the reviewer for this important point. We chose robotics navigation **(Safety-Gymnasium, Section 5, line 266)** as our testbed precisely because its dynamic, sparse-reward nature is a challenging proving ground. However, our framework is not limited to this domain. The core requirement is not LiDAR, but the ability to decompose the state $s_t$ into a graph of interacting entities.
>
> To clarify this and directly address the concern, we have added **Appendix A.15 (Page 41)**. This new section explicitly details how our method generalizes to other modalities, such as using a perception frontend for raw pixel inputs **(lines 2187-2199)** or constructing graphs from discrete cells in tabular MDPs **(lines 2200-2207)**. We further provide concrete formulations for new domains, like Antmaze and Half-Cheetah, in **Appendix A.16 (Pages 41-42)**.
>
> The reviewer is correct that our method requires a state representation with localized, categorized objects. This is a prerequisite of the input (standard in many RL benchmarks), not a limitation of our method's logic. In our paper, the environment's LiDAR vector $L_{obs}(t)$ provides this information directly, as detailed in **Appendix A.2.2 (Page 23)**. Our **Algorithm 1 (Page 26)** operates on this given state. As we now discuss in **Appendix A.15**, this is conceptually equivalent to using a standard object detector as a pre-processing step in a vision-based domain.
>
> Finally, we emphasize that our use of this local, egocentric information is a deliberate feature, as it is far more realistic than assuming global state access. We have expanded on the real-world practicality of this sensor-based approach in **Appendix A.2.2 (lines 1197-1209)**, connecting it to production robotics systems and other major benchmarks. As a direct example - we can consider the largest academic and commercial application - self driving cars. The car (agent) can only observe the vicinity via sensory and other data forms, and one can not assume global state access amidst all the complex interactions in the environment.
>
> ## Response to Weakness 2: Explanation of Graph Construction and Effective Resistance
> ### Reviewer's Comment
> "I would appreciate more explanation on how the graph is constructed and what reducing effective resistance would bring on the main text, but overall the paper is easy to follow."
> ### Our Response
> We are glad the reviewer found the paper easy to follow and appreciate the request for clarification.
>
> Regarding graph construction, the high-level methodology is introduced in **Section 3.2 (Page 3, lines 157-161)**. This section points to the complete, step-by-step procedure in **Algorithm 1 (Page 26)**. Furthermore, a detailed rationale for our specific design choices (e.g., node clustering, connectivity patterns) is provided in **Appendix A.5.1 (Page 28)**. To make this information easier to find, **we have also added additional content to the main text (Section 3.2, lines 171-175) that explicitly directs the reader to these detailed appendix sections for the design rationale, analytical derivations, and empirical validation of our graph construction.**
>
> Regarding the benefit of reducing effective resistance, this is indeed the core intuition of our work. We first introduce it in **Section 1 (Page 2, lines 60-62)**, stating that a decrease in $R_{eff}$ "signifies that the goal has become more structurally accessible." We provide a clear visual intuition for this in **Figure 1 (Page 2)**, which shows how $\mathcal{R}_{eff}$ correctly identifies the more accessible path even when Euclidean distance fails. Finally, we formalize this intuition with a theoretical guarantee in **Lemma 1 (Page 4, line 199)**, proving that decreasing effective resistance is mathematically linked to increasing graph connectivity.
>
> In summary, $\mathcal{R}_{\text{eff}}$ serves as a dense, robust, and theoretically-grounded proxy for "goal accessibility" that holistically accounts for all obstacles and pathways, unlike simpler metrics.

---

> ### Author Response · Authors · 2025-11-19
> **Response to Reviewer FfQG (2/2)**
>
> ## Response to Question: Generality to other MDPs
> ### Reviewer's Comment
> "To increase the impact of this paper, can this method apply to a more general MDP setting, e.g. continuous MDP or tabular MDP, how would you construct the graph in such MDPs? specifically, what are nodes/edges/weights in these settings?"
> ### Our Response
> This is an excellent question that gets to the heart of our method's generality. We first note that our method is already applied to a challenging continuous MDP. The Safety-Gymnasium environments feature high-dimensional **continuous state spaces (e.g., $s_t \in [0, 1]^{128}$ in Appendix A.2.2, Page 23) and continuous action spaces ($a_t \in \mathbb{R}^{\hat{N}}$ in Section 3.1, line 131)**. Our framework is explicitly designed to convert this high-dimensional continuous state $s_t$ into a discrete graph $\mathcal{G}_t$ at each timestep.
>
> The core principle is that our method is applicable to any MDP where the state $s_t$ can be meaningfully parsed into a graph of interacting entities. To thoroughly answer this and demonstrate the framework's flexibility, we have added **Appendix A.15 (Page 41)** and **Appendix A.16 (Page 42)** to the revised manuscript.
>
> As we detail in **Appendix A.15 (lines 2200-2207)**, for a tabular MDP (e.g., a grid world), the nodes would be the agent, goal, and obstacle cells, with edges representing physical adjacency. $\mathcal{R}_{\text{eff}}$ would then serve as a robust, non-Euclidean distance metric that accounts for all walls and pathways. For other continuous settings like raw pixel inputs, **Appendix A.15 (lines 2187-2199)** explains how a standard perception frontend (e.g., an object detector) would extract entities to serve as nodes for our graph construction. We further provide a concrete formulation for complex locomotion tasks (e.g., Half-Cheetah) in **Appendix A.16 (lines 2230-2247)**, showing how body parts can be used as nodes and $\mathcal{R}_{\text{eff}}$ can act as a holistic descriptor of the agent's pose.
>
> We are confident this principle of using $\mathcal{R}_{\text{eff}}$ as a structural, goal-directed reward is broadly applicable, and we are grateful for the opportunity to clarify this.
>
> ### References
> [1] Mees, O., Hermann, L., Rosete-Beas, E., Burgard, W. “CALVIN: A Benchmark for Language-Conditioned Policy Learning for Long-Horizon Robot Manipulation Tasks,” 2022.
>
> [2] Liu, B., Zhu, Y., Gao, C., Feng, Y., Liu, Q., Zhu, Y.,  Stone, P. “LIBERO: Benchmarking Knowledge Transfer for Lifelong Robot Learning,” 2023.

---

> > ### Author Response · Authors · 2025-11-28
> > **Request for Response of the Rebuttal**
> >
> > We hope the reviewer is doing well!
> > We hope that we have addressed the reviewer's concerns appropriately and we hope that the reviewer will increase their score if they are satisfied with the responses.

---

### Author Response · Authors · 2025-11-19
**General Response to Reviewers**

# Common Points and Revisions

We sincerely thank all reviewers for their constructive feedback, which has helped us substantially strengthen our paper. We have incorporated all the revisions, including new SOTA baselines and new appendix sections, to directly address the common points. We actively invite the reviewers to engage in further discussion during the rebuttal period, should any points require further clarification. If you find that our detailed responses, new theoretical insights, and the substantial new empirical evidence provided have satisfactorily addressed your concerns, we respectfully request that you consider increasing your score to reflect the strengthened contribution of this paper. (**Please note that all revisions and additions made to the manuscript PDF are highlighted in blue for easy tracking.**)

To address the comparison, we have now fully integrated two new SOTA methods, NGU and AIM, into our study. Their implementation details are defined in **Section 5 (Page 6, lines 296-304)**. More importantly, we have re-run our experiments and updated all main results to include a direct comparison across all 9 environments. These new results are presented in the updated **Figures 2, 3, and 4 (Pages 7-8)**, where our method's significant performance gains are shown to hold, and in many cases widen, against these strong, recent baselines. For instance, the success rates are over **3x** and **4x** for the highest difficulty level for navigation and building environments respectively. Similar observations hold for the other metrics.

We also appreciate the reviewers' questions on the flexibility and generalization of our framework. To explicitly address this, we have added two new sections to the appendix. **Appendix A.15 (Page 41)** discusses generalization to arbitrary inputs like vision **(lines 2187-2199)**. In more generality, **Appendix A.16 (Pages 41-42)** provides concrete formulations for applying our $R_{eff}$ reward to new domains, where we provide a general graph construction and computation of the effective resistance to construct the intrinsic reward. We then specify details on how to construct graphs for maze environments like Antmaze **(lines 2221-2227)** and for locomotion tasks like Half-Cheetah **(lines 2230-2247)**, showing how $\mathcal{R}_{\text{eff}}$ can serve as a holistic pose descriptor for learning stable gaits.

Finally, we respectfully and strongly argue that our experimental setup, far from being niche, is grounded in real-world practicality. As we have now detailed in the expanded **Appendix A.2.2 (Page 23, lines 1197-1209)**, our use of LiDAR mirrors its primary role ranging from production robotics systems (e.g., Waymo, Boston Dynamics) to most widely used academic benchmarks (e.g., CALVIN [1], LIBERO[2]).

This setup also allows us to clarify a core design choice. Our $\mathcal{R}_{\text{eff}}$ reward is deliberately formulated as a dense, goal-directed local planning signal that activates once the goal is perceived. This complements the global exploration (i.e., finding the goal) driven by the base PPO algorithm and the large $\pm\beta$ reward **(Equation 3, Page 4)**. The success of this decoupled mechanism is empirically proven in **Table 5 (Page 36)**, which shows our final trained policies maintain >98.9% goal visibility across all environments, confirming they successfully learn to both find the goal and robustly navigate toward it.

**References**
[1] Mees, O., Hermann, L., Rosete-Beas, E., Burgard, W. “CALVIN: A Benchmark for Language-Conditioned Policy Learning for Long-Horizon Robot Manipulation Tasks,” 2022.

[2] Liu, B., Zhu, Y., Gao, C., Feng, Y., Liu, Q., Zhu, Y.,  Stone, P. “LIBERO: Benchmarking Knowledge Transfer for Lifelong Robot Learning,” 2023.

---

### Meta-Review · Area_Chair_eHiJ · 2025-12-24

**Summary:**

Across the reviews, the main concerns were generality and positioning (whether the approach is specific to robotics/LiDAR and how to instantiate the graph in more general MDP settings), clarity (how the graph is constructed and intuition for why minimizing effective resistance helps), and experimental completeness (missing comparisons to more recent intrinsic-motivation / GCRL exploration baselines, plus questions about computational overhead and hyperparameter sensitivity).

In rebuttal and the revised manuscript, the authors directly addressed these by (i) adding explicit formulations/instantiations beyond LiDAR (pixels, tabular/grid, locomotion), (ii) adding requested/modern baselines (notably AIM and NGU) into the main experimental plots, and (iii) adding complexity/runtime and robustness discussions.

Given (a) the novelty + theory were not substantively challenged, (b) the empirical story was strengthened with the added baselines, and (c) at least one initially negative reviewer explicitly indicated satisfaction and intent to raise their score, I recommend accept.

**Reviewer Concerns:**

Concerns addressed by the rebuttal / revision

•	Generality beyond robotics/LiDAR: Reviewers asked whether the method is domain-specific and how to define nodes/edges/weights in other MDPs. The authors added new appendices describing extensions to pixels, tabular/grid worlds, and locomotion tasks, and clarified that the core requirement is “state decomposable into interacting entities,” not LiDAR per se.

•	Missing/dated baselines: Requests to compare against stronger or more recent intrinsic motivation / exploration methods (e.g., NGU, AIM) were addressed by adding NGU and AIM and updating the main result plots accordingly.


•	Computational cost & practicality: Questions about complexity and runtime were addressed with added analysis and runtime measurements.

•	Robustness / hyperparameters: The rebuttal argues parameters are theoretically grounded and fixed globally across all 9 environments (no per-environment tuning), and adds text to emphasize this.


Concerns still outstanding (or only partially addressed)

•	Empirical generalization beyond the robotics testbed: While the paper now includes formulations for pixels/tabular/locomotion, the rebuttal evidence for “generality” is primarily conceptual/appendix-based rather than demonstrated by full non-robotics experimental results.

•	Scalability relies on bounded graph size: The efficiency argument depends on keeping the number of nodes small via the construction/abstractions; this is reasonable for the tested regimes but leaves open how it behaves in settings where “entity counts” grow large or are hard to segment reliably.

**Reviewer Scores:**

Reviewer FfQG (Rating 8: accept, good paper): Likely stays at 8, since their main requests were clearer graph-construction explanation and general-MDP applicability, both explicitly addressed via added sections/appendices and revisions.


Reviewer Pwps (Rating 4: borderline): Their core concerns (modality dependence, missing recent baselines, and computational cost/hyperparameter sensitivity) were materially addressed: NGU/AIM were added to the main plots, and runtime/complexity + robustness claims were added.
My estimate if fully engaged: 4 → 6 (weak accept).


Reviewer nmWg (Rating 2: reject): This reviewer explicitly states they are satisfied with the clarifications and additions and will raise their score (pending others’ opinions). They also specifically endorsed the added quasimetric-vs-analytic discussion and only suggested minor stylistic improvements.
My estimate if fully engaged: 2 → 6.

---

### Decision · Program_Chairs · 2026-01-26

Accept (Poster)